# Collaborative QA using Interacting LLMs. Impact of Network Structure, Node Capability and Distributed Data.

**Adit Jain**[*]                                                                aditjain1980@gmail.com

**Vikram Krishnamurthy**                                                         vikramk@cornell.edu

**Yiming Zhang**                                                                 yz2926@cornell.edu

Reviewed on OpenReview: https://openreview.net/forum?id=nyZ4JMrV8b

## Abstract

In this paper, we model and analyze how a network of interacting LLMs performs *collaborative question-answering (CQA)* in order to estimate a ground truth given a distributed set of documents. This problem is interesting because LLMs often hallucinate when direct evidence to answer a question is lacking, and these effects become more pronounced in a network of interacting LLMs. The hallucination spreads, causing previously accurate LLMs to hallucinate. We study interacting LLMs and their hallucination by combining novel ideas of mean-field dynamics (MFD) from network science and the randomized utility model from economics to construct a useful generative model. We model the LLM with a latent state that indicates if it is truthful or not with respect to the ground truth, and extend a tractable analytical model considering an MFD to model the diffusion of information in a directed network of LLMs. To specify the probabilities that govern the dynamics of the MFD, we propose a randomized utility model. For a network of LLMs, where each LLM has two possible latent states, we posit sufficient conditions for the existence and uniqueness of a fixed point and analyze the behavior of the fixed point in terms of the incentive (e.g., test-time compute) given to individual LLMs. We experimentally study and analyze the behavior of a network of 100 open-source LLMs with respect to data heterogeneity, node capability, network structure, and sensitivity to framing on multiple semi-synthetic datasets.

## 1 Introduction

In May 2025, roughly 50% of internet articles were generated using the help of large language models (LLMs), up from 20% in May 2023 according to Paredes et al.. The explosion of LLM-generated text leads to this text being used for training LLMs or using it as their context. Therefore, LLMs (explicitly or implicitly) interact with each other to generate content. Further LLMs have demonstrated improved performance when collaborating with each other in a network-like structure with other LLMs for question-answering, programming, and scientific research (Mitchener et al., 2025). Given the quirks of LLMs (e.g., hallucination, sycophancy), it is crucial to investigate their emergent behavior when interacting with one another.

In this paper, we model and analyze how a network of interacting LLMs performs *collaborative question-answering (CQA)* in order to estimate a ground truth given a distributed set of documents. These distributed documents constitute inputs (referred to as 'context') to individual LLMs. In estimating the ground truth, the LLMs are prone to hallucination [1] when they are provided with a limited non-informative context. Further, in a network of interacting LLMs, hallucination can be either amplified or mitigated depending on the network

---

[*]Adit Jain is currently a Research Scientist at CollinearAI. This project was done during his PhD at Cornell. Vikram Krishnamurthy, and Yiming Zhang are affiliated with the Department of Electrical and Computer Engineering, Cornell University, Ithaca, NY 14853. This work was supported by NSF under grant CCF-2312198.

[1]We define hallucination as reporting a state estimate which is not substantiated by the context and is not the ground truth.

structure. We analyze the spread of information between the LLMs with respect to *network structure, LLM capability* and *distributed data* given their salient features including limited context window and hallucination. To achieve this, we study interacting LLMs by combining novel ideas of mean-field dynamics from network science and the randomized utility model from economics to construct a useful generative model. Figure 1 illustrates our theoretical approach, which we complemented with experimental results on networks of LLMs performing CQA on semi-synthetic real-world datasets.

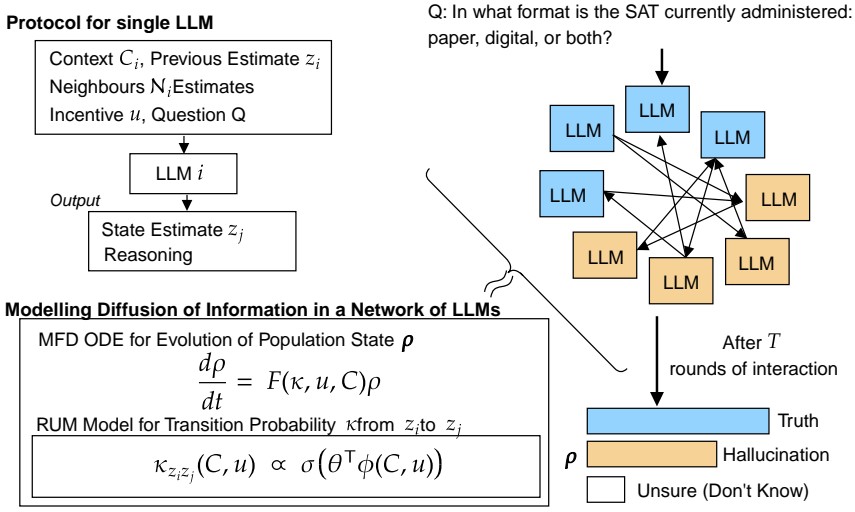

Figure 1: This paper proposes an analytical model for a network of interacting LLMs performing state estimation - we model the information diffusion in the network using a mean-field dynamics (MFD) for directed networks and model the utility of an LLM as being sampled from a random utility model (RUM), which allows for a transition model which can be plugged into the MFD. We further empirically analyze the behavior of a network of 100 open-source interacting LLMs for CQA on different semi-synthetic datasets.

## 1.1 Main Results and Insights for interacting LLMs performing CollaborativeQA

**1. Mean-Field Dynamics for Information Diffusion in a Network of LLMs.** We study the mean-field dynamics, which a generative model where each LLM is represented as an agent with an estimate of the true underlying state, which evolves over time based on its local context and incentives to the case of directed networks. The agent uses its private observation, as well as information from its neighbors, to update its belief about the true state of the world and output a response. To handle the combinatorial complexity of modeling information diffusion in large networks, we derive an ordinary differential equation (ODE) that approximates the average behavior of agents using a mean-field approach in a directed network. We theoretically and experimentally demonstrate the analytical and predictive capabilities of this model.

**2. Randomized Utility Model for Decision Making in LLMs.** To specify the probabilities that govern the MFD, we propose a randomized utility model (RUM) to parameterize the choice probabilities of the LLMs. Such a representation enables us to model contextual information, including the authenticity of sources and the consensus formed by the neighbors of the LLMs. The RUM model provides interpretability on how sensitive LLMs are to changes in their beliefs. The parameters of the RUM can be estimated efficiently using a logistic regression. RUM from economics, operating at a lower level of abstraction, combines seamlessly with the mean-field dynamics from network science, operating at a higher level of abstraction, and provides a useful generative model for decision-making among interacting networked LLMs.

**3. Empirical Study on Semi-Synthetic and Real Datasets Characterizing Interesting Behavior.**
Further we experimentally study information diffusion in network of 100 interacting LLMs (e.g. network of 100 LLaMa3-8B initialized with a power law degree distribution) on three widely used datasets, including Fiction Dataset (which constructed using 30 books of Project Gutenberg similar to the NarrativeXL

dataset Moskvichev and Mai (2023)), Knowledge Cutoff (which we curate using updates on Wikipedia articles), and event-based QA benchmarks (created using news articles from BBC, Reuters and CNN).

As analysts with knowledge of the ground truth, we can classify LLMs into three states: hallucination (H), truthful (T), and don't know (D). Let $\boldsymbol{\rho}_T$ be the fraction of LLMs that are in a truthful state after $L$ interactions, and we refer to this as the truthful population state. We derive the following insights:

> *Insight 1:* The truthful population state $\boldsymbol{\rho}_T$ in a network of LLMs is proportional to the computation used by LLMs (test-time compute) and the base model capabilities of the individual LLMs. This insight is empirically demonstrated in Experiment 1 (Table 1) and Experiment 2 (Figure 4) in Section 3.2.

> *Insight 2:* The placement of the different types of data (correct, missing, incorrect) in a directed network of LLMs affects the $\boldsymbol{\rho}_T$, which increases when more influential nodes have correct data as shown in Experiment 3 (Figure 7) in Section 3.2.

> *Insight 3:* The network structure affects the truthful population state $\boldsymbol{\rho}_T$. If the network is initialized using a power-law degree distribution, then the power law constant affects the extremity as shown in Experiment 6 (Figure 7) in Section 3.2.

The above insights open a new line of research and are instructive for designing networks of interacting LLMs.

## 1.2 Motivation.

*Networks of LLMs as a primitive for Collective Intelligence.* Networks of LLMs now show up in real-world systems and are an example of collective intelligence. Human interactions lead to an emergent collective intelligent sensing behavior that individuals can not (Kraft et al., 2015). Similarly, ensembling in machine learning, and mixture of expert architectures in LLMs, where queries are routed to sub-networks within an LLM are examples of collective intelligence. Since LLMs are trained to analyze and generate human-readable text, they are capable of communicating with other LLMs. Many existing frameworks utilize this approach to design a network of LLM agents that perform specific tasks, such as programming or research. However, little is known about how information propagates in a network of LLMs, specifically in the application of CQA, since LLMs often hallucinate information in the presence of incomplete context or outdated training corpus.

*Network of LLMs in comparison to networks of humans.* Networks of LLMs have two key distinctions from human networks that make their study of scientific interest. Networks of LLMs can be engineered in an isolated environment where the exact context that they use and how they interact can be controlled, therefore allowing for experimentation and exact characterization of their behavior. Secondly, a primary application of a network of LLMs is crowd-sourcing information across a variety of different contexts, and the order of information processing can be arbitrarily scaled with compute. One can choose the computational capability of an LLM to be much larger than that of an average human.

*Motivation for Collaborative QA (CQA). Long-Context:* In the past few years, LLMs have been adopted for different applications, including personal assistants, multimedia analysis, and automating workflows. However, in industrial applications, LLMs still struggle with processing long-context documents reliably without hallucinating facts (Li et al., 2025; Levy et al., 2024). One technique used in industry is retrieval augmented generation, where documents are chunked and then a smaller context is retrieved using the relevant set of documents through a vector search. However, such methods are prone to missing out on key context (Barnett et al., 2024), and approaches like the chain of agents propose performing inference separately on each of the paragraphs (Zhang et al., 2024a). For medical and legal question answering, long-context is often unavoidable Zhang et al. (2025). Further, decentralized LLM networks improve fault tolerance—by avoiding single points of failure, the system becomes more robust to outages or adversarial compromises of individual agents. *Privacy:* Centralized LLMs typically require unrestricted access to complete raw datasets, posing significant privacy concerns in sensitive fields like healthcare and finance Song et al. (2024).

In contrast, networks of LLMs support distributed reasoning, where only intermediate inferences, rather than raw inputs, are exchanged. This preserves data sovereignty and control. Additionally, decentralized architectures offer greater deniability and privacy through fragmentation, since no single agent sees the full input space. This is particularly valuable in regulated domains like healthcare, finance, or defense, where data cannot be pooled due to legal constraints (Peris et al., 2023).

### 1.3   Related Work

**Collaborative Multi-LLM Setups**   Early efforts at multi-agent LLM systems show that coordinating several LLMs can substantially extend the reasoning horizon of a single model. Frameworks such as CAMEL's role-play dialogue between agents (Li et al., 2023), Chain-of-Agents for sequential long-context processing (Zhang et al., 2024b), and Tree-of-Thoughts for parallel search over reasoning paths (Yao et al., 2023) demonstrate accuracy gains on complex tasks, while self-consistency ensembles reduce failure modes (Wang et al., 2023). Larger "societies" of agents, e.g. Generative Agents sandbox Park et al. (2023) of 25 autonomous characters show scalability of natural-language communication for distributed inference. Recent work exposes the fragility of LLM networks to misinformation. Multi-LLM debates converge on shared hallucinations (Estornell and Liu, 2024), and stronger but less truthful debaters can sway both models and humans (Khan et al., 2024; Agarwal and Khanna, 2025). To counter this, researchers borrow ideas from economic mechanism design: peer-prediction style incentives reward truthful reporting even without ground truth (Kong and Schoenebeck, 2019), while the MFD used in this paper offers tractable models for populations of interacting agents (Yang et al., 2018). The work whose framework we extend is Jain et al. (2025), where they consider preferential attachment in a network of LLMs and propose a similar ODE model however their model makes assumptions on the proportion of hallucinating and truthful nodes for all in-degree distribution (A3 in Jain et al. (2025)) which we are able to remove using a more involved expression for the proportion of truthful nodes (See Equation (3)). Further, we conduct a thorough empirical investigation on three different datasets for a variety of different confounders, including topology, communication pattern, and heterogeneity.

Further, there have been many different approaches proposed to make collaboration possible in multi-LLM systems. Recently, Chen et al. (2024a) proposes a technique to generate LLM agents on the fly based on the sub-tasks. AgentsNet is another technique for coordination and collaborative reasoning in LLMs, enabling strategies to be formed through interaction with each other for problem-solving, self-organization, and effective communication, given a network topology (Grötschla et al., 2025). (Qiu et al., 2024) presents a framework for training large language models (LLMs) as collaborative agents to enable coordinated behaviors in cooperative MARL. However, most of these are system-level engineering frameworks, which are very useful in practice but offer little insight into the behavior of LLMs when they interact in a network. Although the main output variable of interest in our study is the percentage of nodes hallucinating, we do not claim novelty in detecting hallucination; rather, for us, hallucination serves as an analytical tool, given that we know the ground truth. There is more extensive discussion of related work in Appendix A.

## 2   Modeling Information Diffusion and Collaborative QA in Interacting LLMs

In this section, we formalize the protocol of interaction for a network of LLMs and then model the spread of information by a mean-field dynamics ordinary differential equation, a deterministic dynamical equation for the evolution of the proportion of LLMs in different information states (which is otherwise stochastic). Furthermore, we propose a randomized utility model, a stochastic choice model that estimates the transition probabilities of an LLM between different states, which can be integrated into the ODE to obtain a predictive model. We demonstrate the utility of this mathematical model by analyzing the existence and behavior of the fixed point of the dynamical system and empirically demonstrate its predictive capabilities. Note that we model collaborative QA (CQA) under the umbrella of information diffusion, which is consistent with network science (Jackson and Lopez-Pintado, 2013), where spread of discrete ideas (e.g., adoption of a product) is studied through the same lens and is similar to modeling spread of epidemics.

## 2.1 Network of Large Language Models performing state estimation for CollaborativeQA

*Network Structure for Interaction:* Let $\mathcal{G} = (\mathcal{V}, \mathcal{E})$ denote a network of $N$ LLMs [2] where $\mathcal{V} = \{1, 2, \ldots, N\}$ denote the vertices, each of which is an LLM and $\mathcal{E} \subseteq \mathcal{V} \times \mathcal{V}$ denote the set of edges between the LLMs. If $(i, j) \in \mathcal{E}$, then there is an edge from $j$ to $i$ and LLM $i$ is influenced by $j$, i.e., it considers the opinion of $j$ before providing an estimate. The set of nodes $i$ influenced by is denoted by $\mathcal{N}(i) = \{j | (i, j) \in \mathcal{E}\}$ and is referred to as the neighbors of $i$, and the in-degree of node is the cardinality of this set. We assume that the LLM network is sampled from a network with joint distribution of in-degree $l$ and out-degrees $m$ denote by $Q(l, m)$. Denote $Q(l|m)$ as the conditional in-degree distribution given the out-degree of the node is $m$.

*Aim:* The aim of the network of LLMs is to estimate the underlying state $x \in \mathcal{X}$, where $\mathcal{X}$ is the state space. This could be an answer to a question (as in the case of CQA) or a fact based on a knowledge base.

*Dynamics of Interaction:* The LLMs interact in a sequential manner in the following fashion: At time $k = 1, 2, \ldots$ one node $i$ is sampled and interacts with its neighbors by receiving their previous state estimate which it then uses to produce an estimate of its own. Note that there are two simplifying assumptions we make here: one is the sequential nature of the interactions, which is not impractical if one notes that the actual time intervals can be arbitrarily close. In any practical system, one usually has some syncing mechanism with resolution for conflicts based on the time stamp when the interaction happens. However, we do large-scale experiments where multiple LLMs interact in parallel at a time.

*Control:* The network of LLMs is also given an incentive or control (or model capability) $u$ from the space $\mathcal{U}$. This incentive could be either an explicit payment to perform a task (for an external LLM-based agent) or in the form of a system prompt that requires more tokens and incurs a higher cost to the learner for inference.

*Signals from the state:* Each LLM $i$ has a private observation $y_i$ (not shared with other LLMs) from the observation space $\mathcal{Y}$ which is part of its context. Each LLM produces a state estimate along with additional output, like a rationale for the decision. To provide a state estimate, an LLM $i$ processes its context, including the previous estimates of its neighbors. The state estimate is denoted by $\hat{x} \in \bar{\mathcal{X}} = \mathcal{X} \cup \{\text{'Dont know'}\}$.

## 2.2 Mean Field Dynamics (MFD) for a Network of LLMs for Population State Evolution

We now discuss an MFD model for information diffusion in a directed network of large language models, and MFD is a generative model for the behavior of a large network of LLMs.

*Motivation for using MFD to model the information diffusion in network of LLMs.* There are three primary motivating factor for our choice of MFD as a model, which align with the of this model in information diffusion in the network science literature: (A) Combinatorial Complexity: An exact dynamical model tracking the joint state of the network would require a state space that grows exponentially with the number of agents ($|\mathcal{X}|^N$). For $N = 100$, this is computationally intractable. The MFD reduces this complexity by approximating the average behavior of the population, making the derivation of the governing ODEs manageable. (B) Analytical Tractability: The MFD allows us to derive closed-form insights regarding the system's convergence. Specifically, it enables us to prove the existence, uniqueness, and global asymptotic stability of the fixed point (Theorem 1). Such analytical characterization of the equilibrium would be difficult to obtain with a high-dimensional exact model. (C) Empirical Validation: In Section 2.6 we empirically validate the approximation for $N = 100$ agents and 1000 sequential interaction rounds, the MFD model accurately predicts the dynamics of the population state with high correlation ($\geq 0.8$) and low KL divergence ($\leq 0.05$).

Denote the current (estimated) state distribution of LLMs with in-degree $l$ by $\boldsymbol{\rho}^l \in \Delta$, where $\Delta$ is the $|\mathcal{X}|$-dimensional simplex. Denote the population state vector as $\boldsymbol{\rho} = (\boldsymbol{\rho}^1, \boldsymbol{\rho}^2, \ldots, \boldsymbol{\rho}^N)$. We use mean-field dynamics and consider that the state distribution evolves with the set of ODEs given by,

$$\frac{d\boldsymbol{\rho}^l}{dt} = \mathbf{F}^l(Q, \boldsymbol{\rho}, u)\boldsymbol{\rho}^l, \tag{1}$$

---

[2]In this paper, we consider a large network of LLMs and assume $N$ to be sufficiently large ($N \gtrsim 100$).

for all degree index $l \in [N]$, for a control $u$ and for a degree distribution $Q$. The net rates of change for the population state of nodes with in-degree $l$ are encoded in the rate matrix $\mathbf{F}^l(Q, \boldsymbol{\rho}, u)$, whose entries are given by,

$$\mathbf{F}^l_{z_1, z_2}(Q, \boldsymbol{\rho}, u) = \begin{cases} G^l_{z_1 z_2}(Q, \boldsymbol{\rho}, u) \geq 0, & z_1 \neq z_2 \quad \text{(inflow from state } z_1 \text{ to } z_2) \\ -\sum_{z'_2 \in \mathcal{Z}, z'_2 \neq z_1} G^l_{z_1 z'_2}(Q, \boldsymbol{\rho}, u) \leq 0, & z_1 = z_2 \quad \text{(total outflow from state } z_1) \end{cases}, z_1, z_2 \in \bar{\mathcal{X}}$$

where $G^l_{z_1 z_2}(Q, \boldsymbol{\rho}, u)$ is the average transition probability given by the following expression,

Note that $\mathbf{F}^l$ is a rate matrix (also known as a generator matrix): its off-diagonal entries are non-negative transition rates, while each diagonal entry is the negative sum of the off-diagonal entries in its row, ensuring that rows sum to zero. Consequently, the diagonal entries are non-positive and represent the net outflow of probability mass from each state. The non-negative transition probabilities themselves are $G^l_{z_1 z_2} \geq 0$ for $z_1 \neq z_2$.

$$G^l_{z_1 z_2}(Q, \boldsymbol{\rho}^l, u) = \sum_{\mathbf{n} \in \mathbb{N}_0^{|\bar{\mathcal{X}}|}, |\mathbf{n}|=l} \kappa_{z_1, z_2}(u, l, \mathbf{n}) \binom{l}{\mathbf{n}} \theta_z(Q, \boldsymbol{\rho})^{\mathbf{n}}, \tag{2}$$

with $\binom{l}{\mathbf{n}} = \frac{l!}{\prod_{z=1}^{|\bar{\mathcal{X}}|} n_z!}, \theta_z(Q, \boldsymbol{\rho})^{\mathbf{n}} = \prod_{z=1}^{|\bar{\mathcal{X}}|} \theta_z(Q, \boldsymbol{\rho})^{n_z}, |\mathbf{n}| = \sum_{z=1}^{|\bar{\mathcal{X}}|} n_z = l$ and $\mathbb{N}_0^{|\bar{\mathcal{X}}|}$ is a $|\mathcal{X}|$ dimensional lattice over whole numbers. $\kappa_{z_1, z_2}(u, l, i, j)$ denotes the probability of a LLM with in-degree $l$ transition from state $z_1$ to state $z_2$ given that it has $i$ truthful and $j$ hallucinating neighbors. $\theta_z(Q, \boldsymbol{\rho})$ denoting the probability that a randomly sampled edge originates from a node in state $z$ is (derivation is given in Appendix B.1),

$$\theta_z(Q, \boldsymbol{\rho}) = \frac{\sum_m \sum_l m Q(l, m) \sum_l \boldsymbol{\rho}^l Q(l|m)}{\sum_m \sum_l m Q(l, m)}. \tag{3}$$

The expression for $G^l_{z_1 z_2}(Q, \boldsymbol{\rho}^l, u)$ in (1) computes the average transition rates for LLMs with a particular in-degree. The key ingredient to instantiating the transition rates in the mean-field ODE of (1) is estimating the transition kernel $\kappa_{z_1, z_2}(u, l, \mathbf{n})$, since these parameters encode how local neighbour configurations and control $u$ drive latent-state transitions in the LLM network. One can estimate the transition kernel using a standard plug-in approach; however, as we explain next, we propose modeling the transition kernel by considering the utilities of the individual LLMs, sampled from a randomized utility model.

### 2.3 Randomized Utility Model (RUM) for estimating the transition probabilities

In order to estimate the transition probabilities of the MFD of (1), we propose using the Randomized Utility Model (RUM). RUM is a macroeconomic probabilistic choice model when the choice is between discrete alternatives. The model was proposed to model and analyze choices of rational agents from a population McFadden (1974). The implicit assumption is that the LLMs act as rational agents that make decisions by maximizing a utility function which is a noisy version of a utility function.

Motivation for RUM to predicting the transition probabilities: While transition probabilities could indeed be estimated via simple counting (plug-in estimates) or black-box neural networks, we specifically chose RUM for two main reasons, (A) Interpretability: RUM grounds the model in decision theory, treating LLMs as "rational" agents that maximize a utility function involving distinct features (e.g., social consensus vs. private context). This allows us to inspect to quantify exactly how much weight an LLM places on its neighbors versus its own hallucination. (B) Analytical Tractability: As demonstrated in Theorem 1, the specific parametric form of the Logit model (derived from RUM under Gumbel noise) satisfies smoothness and monotonicity properties that allow us to mathematically prove the existence and uniqueness of the network's fixed point.

In a population of LLMs we propose that a randomly sampled agent maximizes a random utility $r_{z_2}(u, l, \mathbf{n}, w, z_1)$, where $u \in \mathcal{U}$ is a system-wide control (e.g. extra tokens, tool access, model-capability), $l = |N(i)|$ is the in-degree of $i$, $\mathbf{n}$ is the vector of empirical distribution of neighbor's answers, respectively, $w$

summarizes the textual context provided to the LLM[3], $z_i \in \bar{\mathcal{X}}$ is $i$'s current state estimate. The key idea underpinning RUM is that the realized utility for an LLM for providing state estimate $z$ is corrupted by additive noise (assumed to be a Gumbel distribution),

$$\bar{r}_z(u, l, \mathbf{n}, w, z_1) = \theta^\mathsf{T} \phi_z(u, l, \mathbf{n}, w, z_1) + \varepsilon, \qquad \varepsilon \sim \text{Gumbel}(0, 1), \tag{4}$$

where $\theta \in \mathbb{R}^d$ is an unknown parameter vector, $\mathbf{n} \in [n]^{|\mathcal{X}|}$ is the vector of the counts of the neighbours in different states and $\phi_z(u, l, \mathbf{n}, w, z_1)$ is the feature vector encoding the aspects described above. With the IIA property, one obtains the multinomial logit choice rule (McFadden, 1974). Given $(u, l, \mathbf{n}, w, z_1)$, the probability of transitioning to state $z_2$ is then given by, $\kappa_{z_1, z_2}(u, l, \mathbf{n}, w) = \frac{\exp(r_{z_2}(u, l, \mathbf{n}, w, z_1))}{\sum_{z \in \mathcal{Z}} \exp(r_z(u, l, \mathbf{n}, w, z_1))}$. The Gumbel noise in (**??**) implies the independence-of-irrelevant-alternatives (IIA) property; if empirical tests reject IIA, one can replace the soft-max transition probabilities with mixed or nested logit. A single parameterized utility explains all transition directions and generalises to unseen controls $u$ or neighbor configurations $(i, j)$. Directly fitting $\kappa$ would scale poorly $(O(|\mathcal{U}|L^2))$ and does not provide interpretable insight insight. Estimating the parameters is a logistic regression problem, which we describe in Appendix B.3.

## 2.4 Analyzing effects of different problems using a 2-state version

To analytically derive insights for the fixed point of the MFD ODE, we consider the following simplification: We restrict the MFD model to a two-state system (with only truthful or hallucinating nodes). Under the two-state MFD model and the randomized utility model presented in Section 2.3, one can derive interesting analytical properties of the fixed point of the ODE, as well as the effect of the incentive $u$ on the fixed point, under reasonable assumptions on the utility and transition probabilities. We introduce the variable $q = \frac{m}{l}$. We assume the following,

(A1) (Monotone social influence) $\Delta_H(u, l, q, w) := r_T(u, l, q, w, H) - r_H(u, l, q, w, H)$ and $\Delta_T(u, l, q, w) := r_T(u, l, q, w, T) - r_H(u, l, q, w, T)$ satisfy $\partial_q \Delta_H \geq 0$ and $\partial_q \Delta_T \geq 0$ for all $q \in [0, 1]$ for all $l \in [0, N]$.

(A2) (Smoothness) $S_H := \sup |\partial_q \Delta_H| < \infty$ and $S_T := \sup |\partial_q \Delta_T| < \infty$.

(A3) (Non-degenerate switching) $\eta(u) := \inf_{\theta \in [0,1], l} \left( A_l(\theta; u) + B_l(\theta; u) \right) > 0$.

(A4) (Incentive direction) $\partial_u \Delta_H \geq 0$ and $\partial_u \Delta_T \geq 0$.

**Theorem 1** (Fixed point and comparative statics under state-dependent RUM). *Let $\overline{X} = \{T, H\}$. For a node with in-degree $l$, define the state-conditioned multinomial-logit kernel and let $M \sim \text{Bin}(l, \theta)$ count truthful neighbors when the edge–truth rate is $\theta \in [0, 1]$. Write*

$$A_l(\theta; u) := \mathbb{E}\big[\kappa_{H,T}(u, l, M)\big], \qquad B_l(\theta; u) := \mathbb{E}\big[\kappa_{T,H}(u, l, M)\big], \qquad \rho_l(\theta; u) := \frac{A_l(\theta; u)}{A_l(\theta; u) + B_l(\theta; u)}.$$

*With the joint degree distribution $Q(l, m)$, define the edge-weighted scalar map*

$$\Phi(\theta; u, Q) := \frac{\sum_{l,m} m \, Q(l, m) \, \rho_l(\theta; u)}{\sum_{l,m} m \, Q(l, m)} \in [0, 1].$$

*Under the assumption (A1-A4) Then the following holds true,*

*(i) $\Phi(\cdot; u, Q)$ is continuous and non-decreasing on $[0, 1]$, hence admits a fixed point $\theta^\star \in [0, 1]$.*

*(ii) If $\frac{\max\{S_H, S_T\}}{4\,\eta(u)} < 1$, then $\Phi$ is a contraction on $[0, 1]$. Consequently, the fixed point $\theta^\star$ is unique and globally asymptotically stable for the mean-field dynamics $\dot{\theta} = \Phi(\theta; u, Q) - \theta$.*

---

[3]In practice, $w$ is a learned feature vector extracted from the question and the non-private context of the LLMs.

*(iii) (Comparative statics in u) The fixed point $\theta^\star(u)$ is non-decreasing in $u$. If $\mathcal{U}$ is continuous and compact,*

$$\frac{d\theta^\star}{du} = \frac{\displaystyle\sum_{l,m} m\, Q(l,m)\, \frac{A_{l,u}(\theta^\star; u)\, B_l(\theta^\star; u) + A_l(\theta^\star; u)\, \widetilde{B}_{l,u}(\theta^\star; u)}{\big(A_l(\theta^\star; u) + B_l(\theta^\star; u)\big)^2}}{\displaystyle\sum_{l,m} m\, Q(l,m)} \cdot \frac{1}{1 - \Phi'(\theta^\star; u, Q)},$$

*where $A_{l,u} = \mathbb{E}\big[\sigma'(\Delta_H)\, \partial_u \Delta_H\big]$ and $\widetilde{B}_{l,u} = \mathbb{E}\big[\sigma'(-\Delta_T)\, \partial_u \Delta_T\big]$. Proof is in Appendix B.2.*

*Implication for Practitioner.* Any incentive $u$ that raises the likelihood of truth in either state ($\partial_u \Delta_H, \partial_u \Delta_T \geq 0$) increases the equilibrium truth level $\theta^\star$, with effect sizes largest where decisions are "soft" (through $\sigma'$) and amplified by edge-weighting of high–out-degree nodes. The theorem shows that for each in-degree $l$, the state-dependent RUM logits produce $A_l(\theta; u) = \mathbb{E}[\kappa_{H,T}]$ and $B_l(\theta; u) = \mathbb{E}[\kappa_{T,H}]$, whose ratio $\rho_l = A_l/(A_l + B_l)$ is the steady truthful share at that degree; the global map $\Phi(\theta; u, Q)$ then edge-weights these shares by out-degree (exposure) and a fixed point $\theta^\star = \Phi(\theta^\star; u, Q)$ is a network equilibrium. A1 ensures $\Phi$ is monotone, so a steady state always exists; uniqueness and global convergence follow when the slope bound $\max\{S_H, S_T\}/(4\,\eta(u)) < 1$ holds, which compares the strength of social responsiveness to the amount of flow between the two states.

## 2.5 Discrepancy between sequential MFD and parallel experimentation

While our theoretical derivation relies on sequential updates for analytical tractability, the experimental validation utilizes parallel interactions for computational efficiency. We justify this approximation through the global asymptotic stability established in Theorem 1. Since the basin of attraction for the unique fixed point $\theta^*$ covers the entire valid interval $[0, 1]$, there is no risk of the parallel updates inducing oscillations large enough to 'escape' the stable region, as no such boundary exists within the state space. As long as the contraction condition is met ($max\{S_H, S_T\}/4\eta(u) < 1$), the parallel dynamics are constrained to converge to the same fixed point as the sequential model, differing only in the rate of convergence rather than the final state.

## 2.6 Validating Predictive Capabilities of the Theoretical Framework

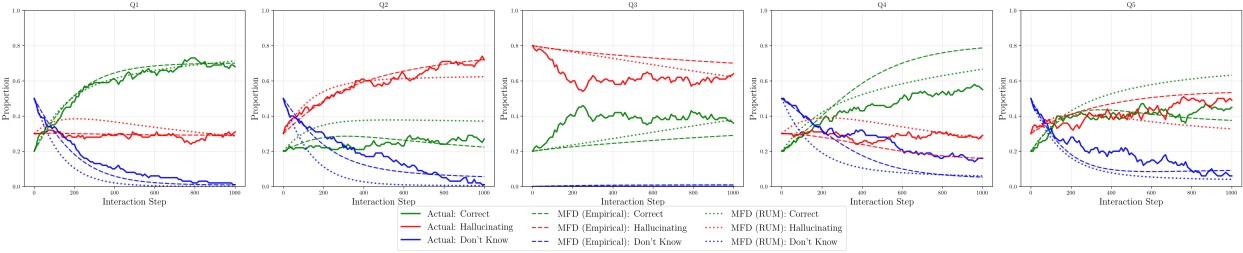

Figure 2: Illustrating the predictive capabilities of the Mean-Field ODE for a network comprising 100 LLMs specified in (1). The mean-field ODE with the RUM model accurately predicts the dynamics of the population state for different questions by estimating the parameters from the first 150 interactions, enabling application of systematic analysis, simulation-based studies, and control theoretic frameworks.

To demonstrate our theoretical framework, we validate the mean-field dynamics on a collection of stylized problems described in Appendix C.2 for a network of $N = 100$ LLM agents, each of which has a different context but the same base model, which is Gemini-2.5-Flash-Lite. We fit an ODE model to the first 150 observations. The fraction of truthful nodes and the prediction of the fitted ODE model in Figure 2. There are two fitted models, (a) without RUM (simple empirical estimates), which has an average trajectory (over 10 runs) correlation of $0.9317 \pm 0.0143$ and an average KL divergence of $0.0441 \pm 0.0550$ (b) with RUM with correlation of $0.8985 \pm 0.0373$ and KL Divergence of $0.0392 \pm 0.0264$. Note that we have primarily highlighted the analytical benefits of using such a model, as our setup requires knowledge of the ground truth to determine whether the LLM was truthful or not. However, one can use the same setup for a standard

categorical variable (MCQ for a question), and use the ODE to predict the future behavior of the network of LLMs. This is used to control them, for example, to prevent the spread of certain sentiments.

In Theorem 1, we show the analytical tractability that the MFD model offers to analyze information diffusion in a network of LLMs with respect to the fixed point of the population state evolution. Further, we empirically validated the predictive capabilities of the theoretical framework on a set of questions. Next, we empirically validate the theoretical results, specifically, the monotonic increase in the truthful population state with increasing incentive. Further, we also empirically show the effect of network structure, data assignment and sensitivity to the prompt.

## 3  Experimental Results On Network of Interacting LLMs.

We proposed an analytically tractable model for information diffusion in LLMs, which can be useful in analyzing convergence and fixed points, as we demonstrate in Theorem 1, and also useful in predicting the behavior empirically. However, there has been little research into how empirical networks of LLMs perform on different QA tasks, especially when the context is very long (e.g., multiple books) or is prone to errors (e.g., news events). Therefore, in this section, we empirically study the effects of different model capabilities, network structures, and data heterogeneity on three different datasets on a network of 100 LLMs.

### 3.1  Experimental setup, dataset and task description

*Base Network Configuration.* Unless otherwise specified, we consider the following experimental setup: We set up a network of 100 LLMs which have the same base model: LLaMa-3.1-8B but differ from each other in the context provided to them. These LLMs communicate over a network initialized whose out-degree distribution is a power-law distribution with constant $\gamma = 2.7$. To be more precise for each node $i$ the out-degree $l$ is sampled as $\mathbb{P}(l) \propto l^{-\gamma}$, then $l$ nodes are sampled which are the out-neighbors of node $i$. For computational tractability, we clip the number of edges to 50. The LLMs interact in parallel for 10 rounds (the mean-field dynamics model is sequential). Each agent is provided with the answer of the previous agents, their previous response and a context as part of the system prompt.

We focused primarily on Power-Law (Scale-Free) networks for two reasons: (a) Real-World Relevance: Power-law distributions most accurately model the "influencer" dynamics present in the real-world social and information networks we aim to simulate (e.g., social media, citation graphs). Our study focuses heavily on how "influential nodes" drive hallucination or truth, making this topology the most critical to analyze. (b) Computational Constraints: Simulating a network of 100 interacting LLMs is computationally intensive. We prioritized the topology that yielded distinct structural insights regarding node capability and influence.

*Task Description: Collaborative QA.* The network of LLMs is tasked with answering a set of questions given a distributed dataset correctly, where each LLM possibly has different contexts (a subset of the distributed dataset). As we describe next, each dataset is divided into correct, incorrect, incomplete or empty context. Given the entire context, each question has a single correct answer and the LLM is given a choice to choose from three different choices, a correct choice, an incorrect choice and dont-know choice. Therefore, hallucination is defined for this experimental setup as: the LLM choosing an incorrect response when given a partial or complete context. For all the experiments, 35% of the LLMs are provided with a correct context and the rest are allocated the context randomly with a uniform distribution. In each round, the LLMs interact and update their estimate. For each experiment, we track the population state $\boldsymbol{\rho} = (\boldsymbol{\rho}_T, \boldsymbol{\rho}_H, \boldsymbol{\rho}_D)$ which specifies the proportion of LLMs in different states (truthful, hallucinating, and dont-know).

For the experiments 1 and 2, we consider two different controls $u$, the length of the communication between the nodes and the test-time compute of individual nodes

All experiments are reproducible, and the code and datasets are available on the following repository.

*Dataset Generation Pipeline: .* We first describe the pipeline used to generate the semi-synthetic (*cutoff*, *event*) and synthetic datasets (*fiction*). We retrieve a long-context text from a source (web, book, Wikipedia, existing dataset). We denote the total number of such long-context texts by $N_{\text{texts}}$. Further, we divide the long-context text into smaller paragraphs, each of which is crafted by concatenating possibly non-contiguous

blocks of text. For each long-context text, we generate $N_C$ such paragraphs each of size 1000 tokens. We then generate question-answer pairs using the long-context text. For each long-context text we generate $N_q$ question-answer pairs. To generate the questions, we first feed the complete text (1500 tokens) to a larger model (GPT-5) and a specific paragraph, and ask to generate questions that are only related to that specific paragraph. We generate the answer for each question and also a hallucinated answer. So there are 3 choices for each question. We then post-process the data by first attributing which paragraph is enough to answer which question correctly and then embellishing or rephrasing some paragraphs. In case some paragraphs are synthetically embellished by us, we still define the truth with respect to the initial text.

*Dataset Description.* We benchmark behavior on three Collaborative QA datasets: *cutoff*, *event* and *fiction*.

*Fiction Dataset.* For the first dataset we take $N_{\text{texts}} = 30$ books from the Gutenberg project similar to Moskvichev and Mai (2023) and then bifurcate each of them into 5 paragraphs each with a few hundred tokens. We then create 5 question-answer pairs each. There is data contamination here since most open and closed source models are trained on books from Project Gutenberg. Note that this still serves as a good dataset to benchmark on since LLMs often fail to retrieve facts that are there in their training data, and so it is interesting to see how truth/hallucination spreads when a few of the LLMs have access to the correct context. We label this dataset as *fiction* dataset. The main purpose of this dataset is to evaluate how the network of LLMs performs on a fictitious fact retrieval task based on the provided context.

*Knowledge Cutoff Dataset.* We create a semi-synthetic dataset based on the cutoff date of LLaMa-3 series model. We use the same pipeline as DatedData (Cheng et al., 2024), which uses the edit history of Wikipedia articles to determine facts that have changed after the cutoff date, which is December 03, 2023. We use LLaMa-3-70b for the hallucinated answer. We obtain the correct answer by updating the wiki page, passing it to the ChatGPT API, and verifying it further using the Perplexity API. We either give the models the correct (updated) answer or not. Question has the correct date. We label this dataset as the cutoff dataset, and the main purpose is to analyze the implicit bias that the LLMs have in hallucinating facts and how it spreads etc to other LLMs.

*Event Description Dataset.* We obtain 100 news articles from April 2025 from Reuters, CNN, and BBC. We use each news article to synthetically generate 5 different styles of narrative - the article itself, an independent journalist, a X Post, a newsletter/essay or a thread of forum (e.g. Reddit). The narratives can include bias, inaccuracies and partial information. We generate one pair of question-answer that can be answered using the original news article, and we also generate a hallucinated fact for that article. We label this as the *event* dataset. The main purpose of this dataset is to see how heterogeneity in framing affect information diffusion.

## 3.2 Experiments

*1. Impact of Different Communication Overhead.* We first analyze the effect of the maximum length of communication (measured in tokens) that the LLMs are allowed to have with each other. Of course, more tokens can carry more information, but they can also potentially help spread lies, etc. This is one example of the control variable $u$ that affects the transition probabilities of (4) and therefore changes the ODE of (1). We observe that the latter is not the case and report our results (the final fraction of truthful LLMs) in Table 1 for the different datasets. It can be seen that although the fraction of truthful LLMs, $\boldsymbol{\rho}^T$ after interaction is increasing in the number of communication overhead, it has diminishing returns. This validates the claim of the truthful proportion increasing with the incentive $u$ from Theorem 1(iii).

*Experiment 2: Impact of different controls under different heterogeneity.* We compare the effect of the level of more sophisticated test-time scaling methods (system prompts or strategies) on the eventual convergence of the fraction of truthful LLMs $\boldsymbol{\rho}^T$ in Figure 3. Specifically, we consider the number of deliberation steps the LLM takes before committing to an answer. This is another example of the control $u$ that can be applied to the network of LLMs. Each deliberation step is a chain of thought followed by an estimate of the answer, and between each deliberation step, there is a self-critique that the LLM does. different controls. It is clear that test-time methods, which improve the performance of a single LLM, also scale well to a network of LLMs. The trend of increasing and concavity in the improvement broadly still holds. We also benchmark this on a network of heterogeneous LLMs where 20% of the nodes are randomly assigned a closed-source

| Dataset | Metric | Length | | |
|---|---|---|---|---|
| | | Answer Only | 50 Tokens | 100 Tokens |
| **Event** | T | $0.589 \pm 0.412$ | $0.623 \pm 0.043$ | $\mathbf{0.656 \pm 0.047}$ |
| | H | $\mathbf{0.411 \pm 0.412}$ | $0.251 \pm 0.041$ | $0.234 \pm 0.045$ |
| | DK | $0.000 \pm 0.000$ | $\mathbf{0.127 \pm 0.035}$ | $0.110 \pm 0.028$ |
| **Fiction** | T | $0.537 \pm 0.415$ | $0.631 \pm 0.047$ | $\mathbf{0.697 \pm 0.046}$ |
| | H | $\mathbf{0.463 \pm 0.415}$ | $0.245 \pm 0.045$ | $0.206 \pm 0.042$ |
| | DK | $0.000 \pm 0.000$ | $\mathbf{0.123 \pm 0.030}$ | $0.098 \pm 0.030$ |
| **Cutoff** | T | $0.458 \pm 0.412$ | $0.510 \pm 0.321$ | $\mathbf{0.539 \pm 0.327}$ |
| | H | $\mathbf{0.542 \pm 0.412}$ | $0.328 \pm 0.219$ | $0.306 \pm 0.220$ |
| | DK | $0.000 \pm 0.000$ | $\mathbf{0.161 \pm 0.108}$ | $0.155 \pm 0.115$ |

Table 1: Proportion of truthful, hallucinating, and don't-know LLMs in the last iterate for different communication overhead (tokens): The proportion of truthful LLMs increases and has diminishing returns with increasing communication overhead.

model (GPT-4.1-mini) in Figure 3, and observe that the trend is the same, but the closed-source network is able to push the proportion of truthful LLMs.

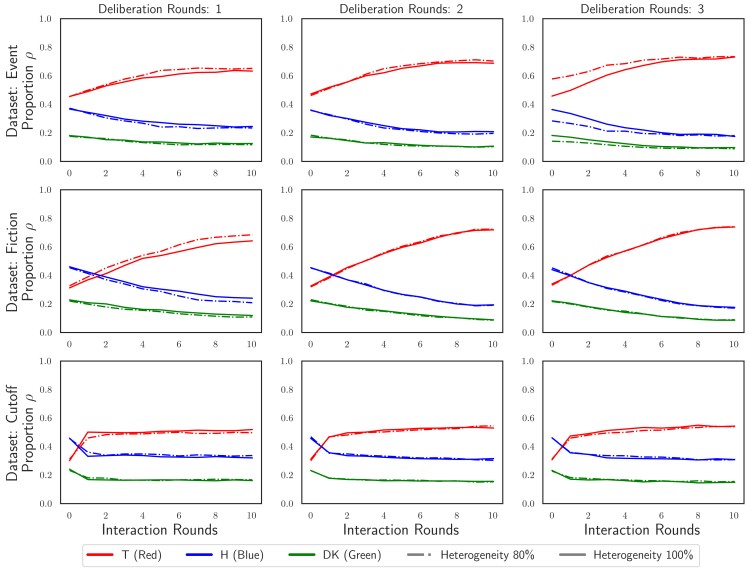

Figure 3: Evolution of population state $\boldsymbol{\rho}$ (vs interaction rounds) for the three CQA datasets with different numbers of deliberation rounds and different levels of heterogeneity in the network. As intuitively expected, increasing the number of deliberation rounds results in a better outcome (fraction of truthful LLMs).

*Experiment 3: Impact of context placement.* We study the impact of providing different (correct or incorrect/incomplete) contexts on influential nodes and plot the results for different topologies for the fiction dataset in Figure 4. Influence here is defined differently for different networks, for chain networks it is the beginning nodes, for the tree network nodes closer to the root are more influential and for power-law distributed degree distribution the nodes with higher degree centrality have higher influence. It can be seen that placing the correct data on influential nodes leads to an improved proportion of truthful nodes, and placing it on less influential nodes does not change the truthful proportion much. Therefore, the placement of the initial context in a network of LLMs plays an important role in deciding the eventual belief of the network.

*Experiment 4: Impact of Model Heterogeneity.* Next, we examine the impact of model heterogeneity, i.e., having models of varying capabilities. We use a 3 billion parameter model and an 8 billion parameter model, and we adjust the node placement to plot the convergence plots in Figure 5. It can be seen that more LLMs converge to the truth if a stronger LLM (8 billion versus 3 billion parameters) has a higher degree centrality.

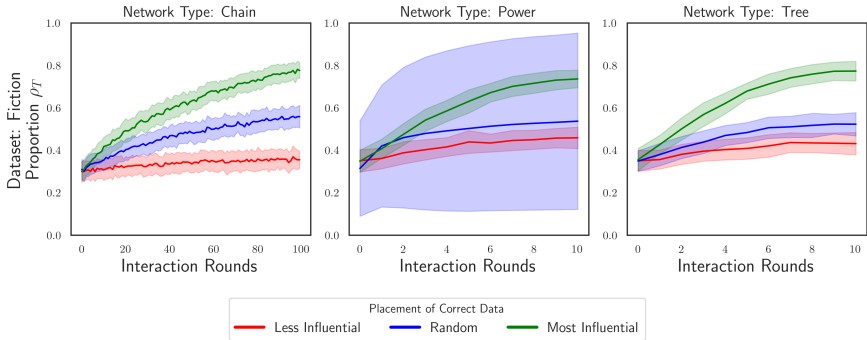

Figure 4: For different networks (*chain*, *power-law* and *tree*), the placement of context affects the fixed point of the population state that the LLM network converges to: if the correct context is placed on more influential nodes the network converges to a higher $\rho_T$ (truthful proportion of LLMs). When the data is randomly assigned to LLMs in a network, then we observe that a network with a power law distribution has higher variability compared to chain and tree network structures.

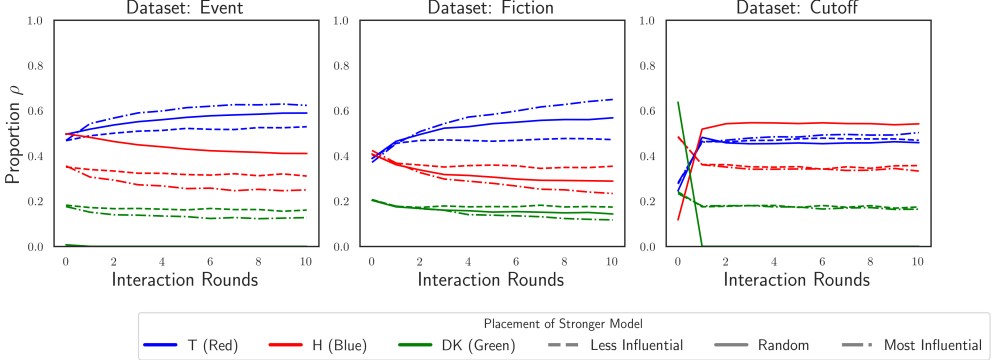

Figure 5: For the three datasets, (*event*, *fiction* and *cutoff*), the strength of the influential node affects the convergence of the population state $\boldsymbol{\rho}$: More LLMs converge to the truth if a stronger LLM (8 billion versus 3 billion parameters) has a higher degree centrality.

*Experiment 5: Impact of the number of LLMs.* The impact of the number of LLMs on the difference in proportion of truthful and hallucinating LLMs is reported on the three datasets in Figure 6. We perform an additional experiment on a subset of size 20 Q/A on the fiction dataset. We assign the correct context to the influential nodes and do 20 independent runs for each Q/A. We report the results in Table 4 in the Appendix.

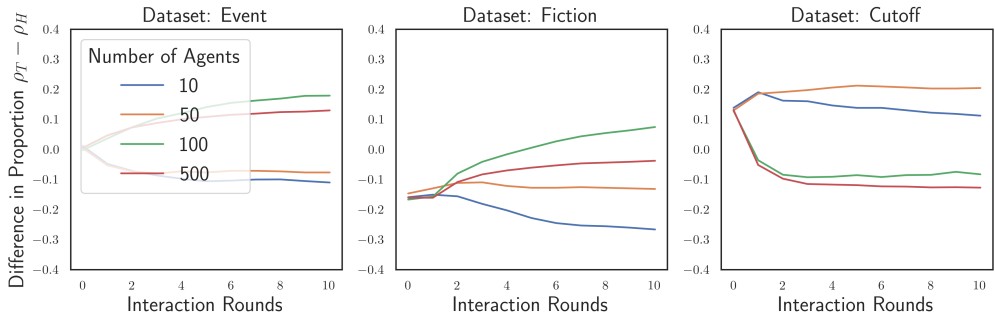

Figure 6: Illustrating the average difference between the evolution of the truthful and hallucinating population state $\rho_T - \rho_H$ for different number of agents.

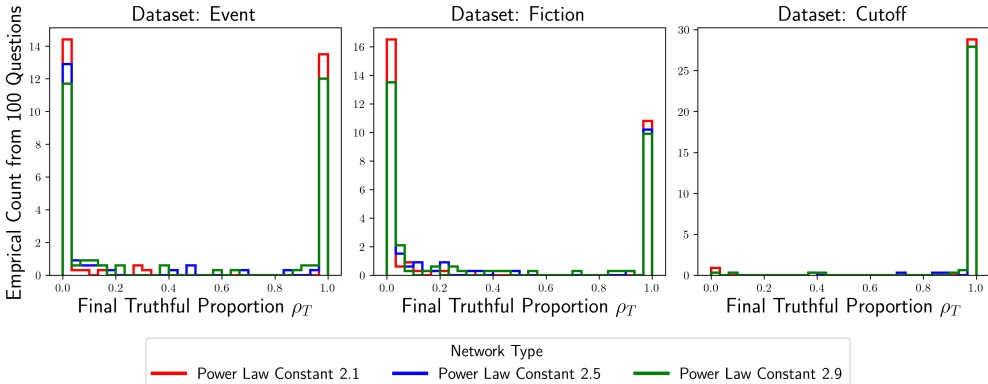

Figure 7: Decreasing the power law exponent (of the degree distribution of LLM network) results in the population state $\boldsymbol{\rho}_T$ becoming more extreme. Hence, if the aim is to make the outcome of the CQA task less extreme, it is necessary to design the LLM network with a higher power law exponent.

| Perturbation | 0 | 1 | 2 | 3 | 4 | 5 | 6 | 7 | 8 | 9 |
|---|---|---|---|---|---|---|---|---|---|---|
| 0 0 0 | 0.96±0.01 | 0.81±0.10 | 0.89±0.05 | **0.97±0.01** | 0.81±0.05 | 0.99±0.01 | 0.99±0.00 | 0.99±0.01 | 0.97±0.01 | 0.96±0.01 |
| 0 0 1 | 0.95±0.03 | 0.95±0.03 | 0.92±0.03 | 0.89±0.01 | 0.88±0.13 | 0.98±0.02 | 1.00±0.00 | 0.96±0.06 | 0.99±0.01 | **0.98±0.01** |
| 0 1 0 | **0.99±0.01** | 0.95±0.01 | 0.82±0.08 | 0.97±0.03 | 0.87±0.05 | 0.99±0.02 | 1.00±0.00 | 0.99±0.01 | 0.98±0.01 | 0.97±0.01 |
| 0 1 1 | 0.98±0.01 | 0.96±0.01 | 0.80±0.04 | 0.96±0.02 | 0.85±0.06 | 0.99±0.01 | 1.00±0.00 | 0.99±0.02 | 0.99±0.01 | 0.98±0.01 |
| 1 0 0 | 0.86±0.12 | **0.97±0.01** | 0.91±0.06 | 0.91±0.03 | 0.79±0.16 | **0.99±0.01** | 0.99±0.01 | 0.99±0.01 | 0.99±0.01 | 0.98±0.02 |
| 1 0 1 | 0.94±0.02 | 0.92±0.07 | 0.91±0.02 | 0.91±0.03 | 0.91±0.04 | 0.99±0.01 | **1.00±0.00** | 0.92±0.04 | **0.99±0.01** | 0.96±0.03 |
| 1 1 0 | 0.73±0.18 | 0.80±0.12 | 0.89±0.05 | 0.88±0.04 | 0.92±0.02 | 0.99±0.01 | **1.00±0.00** | 0.99±0.01 | 0.99±0.00 | 0.97±0.02 |
| 1 1 1 | 0.94±0.03 | 0.66±0.12 | **0.95±0.04** | 0.93±0.01 | **0.93±0.02** | 0.98±0.01 | 1.00±0.01 | **1.00±0.00** | 0.99±0.01 | 0.98±0.02 |

Table 2: The population state $\boldsymbol{\rho}_T$ of the LLM network converges to a fixed point that is robust to framing of the question w.r.t. three perturbations: *lexical paraphrase*, *syntactic re-framing* and *indirect formulation*. The result shows that the population state $\boldsymbol{\rho}_T$ of the LLM network is less sensitive to framing of the question compared to other factors such as dataset placement and network structure.

There are questions (e.g. Q15 and Q17) for which there is no clear trend when increasing the number of LLMs from N=100 to N=500.

*Experiment 6: Impact of different initialization of the network.* Our last experiment examined the different initializations that the network can have using the power law distribution-based random network initialization for different values of the power-law constant. The results of Figure 7 show that decreasing the power law exponent (of the degree distribution of LLM network) results in the population state $\boldsymbol{\rho}_T$ becoming more extreme. Hence, if the aim is to make the outcome of the CQA task less extreme, it is necessary to design the LLM network with a higher power law exponent.

*Experiment 7: Sensitivity.* We perform sensitivity analysis of the convergence of the network of LLMs with respect to framing of the question, and the results are presented in Table 2. We apply 8 different types of linguistic perturbation, which are items from the power set of three operations: *lexical paraphrase*, *syntactic re-framing*, and *indirect formulation* (described in Appendix C.4). We do this for 10 QA pairs from the event dataset. It can be observed that the framing does not have a substantial impact on most questions, and the network of LLMs is generally robust to the framing of the question.

## 4 Conclusion and Future Directions

As intelligent systems like LLMs become more integrated in our society and interact with the content generated by each other, it becomes important to study and characterize the emergent behavior that results from their interaction. This paper studies this theoretically and empirically through the lens of how information propagation in a controlled network of LLMs and the behavior of the fixed point of the information, which is an emergent property. We study the problem of modeling and analyzing a network of interacting large

language models (LLMs), specifically when they have heterogeneity in terms of the context provided to them. We model the interaction in a large number of LLMs using a mean-field dynamics (MFD) for information diffusion over a directed network. Further, to estimate the transition matrices of this mean-field ODE, we propose a randomized utility model (RUM), which models the decision-making of the population and can be estimated in a data-driven way. From a modeling perspective, using RUM for MFD is a novelty that past work had not explored. We present theoretical results which show the properties of the fixed point in a 2 state system. We perform a wide range of controlled experiments to illustrate the different properties of the network of LLMs. Our takeaways reveal that truth propagation scales with model capability, influential node placement, and network topology. Power-law structures and compute-rich agents are most effective in spreading factual consistency across the system.

*Future work.* There are many interesting extensions one can study. (a) One can extend dynamic networks and preferential attachment methods studied in prior work to analyze the *Glass-Ceiling Effect*, wherein influential nodes have privileged information and can influence other smaller models. (b) For knowledge retrieval tasks, one can study the optimal distribution of datasets on a graph and analyze the network from the perspective of *Strength of Weak Ties*. (c) Further, one can study *Communities of LLMs* where the LLMs can collaborate across communities to solve problems. (d) More ambitiously, one can examine the *Incentivization of Agents* and then employ multi-agent reinforcement learning to improve their communication adaptively.

*Broader Impact.* Similar to other research that aimed at reducing undesirable effects like hallucinations, the research presented in this paper can also be used for adversarial or malicious purposes. Particularly, one can optimize the network structure, incentives, and communication protocols to spread hallucination, triggering content, and misinformation. Bad actors can misuse techniques from network science and optimization to engineer echo chambers and amplify certain sentiments, which, paired with sycophancy, can be very detrimental to human-AI interactions on the internet. Therefore, there is more research needed to make LLMs and their networks robust, safe, and reliable. This research also forms a foundation to study network poisoning and analyze how many contaminated nodes are needed to propagate a false belief across the network.

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

## A  Other Related Work

Research has also empirically studied how LLMs can perform collective innovation when they play Little Alchemy 2, a creative video game originally developed for humans that, as the authors argue captures useful aspects of innovation landscapes (Nisioti et al., 2024). They study groups of LLMs that interact with each other about their behavior and show the effect of social connectivity on collective performance. Zhuge et al. (2024) proposes GPTSwarm, a computational graph-based approach for LLM collaboration where each LLM implements a function to process data or query LLMs, and the edges describe the information flow between operations.

Emergent social learning in multi-agent reinforcement learning systems has been studied in the past (Ndousse et al., 2021) and more recently Jain and Krishnamurthy (2025) studies Bayesian learning in a sequence of LLM agents acting as sensors on a stream of data. Information diffusion can be seen as a form of social learning; however, we consider more general graph structures (than line graphs studied in Jain and Krishnamurthy (2025)) and propose analytical models for the average behavior of a large network of LLMs. Yan et al. (2025) serves as a good communication-centric survey on LLM-based multi-agent systems, where they examine key system-level features such as architecture design and communication goals, as well as internal mechanisms like communication strategies, paradigms, objects, and content. (Tran et al., 2025) also serves as a good survey paper on multi-agent collaboration mechanisms.

*Simulating Human Behavior through LLM Networks:*  There has also been research which uses networks of LLMs to simulate human behavior, examples of which include Network of Economic Agents  (Karten et al., 2025) and a mean-field based framework useful for simulating population decision dynamics (Mi et al., 2025). Lu et al. (2024) advocates for considering LLM networks as a tool for complex systems research.

**Elliciting Truthful Behavior**    Recently Peer Elicitation Games (PEG) was proposed as a training-free, game-theoretic framework for aligning LLMs through a peer elicitation mechanism involving a generator and multiple discriminators instantiated from distinct base models (Chen et al., 2025). As a network of LLMs becomes more prominent, such protocols can perhaps be treated as prerequisites to join the network. Further, (Chen et al., 2024b) prompts LLMs to play different roles in a problem-solving team, and encourages different role-play agents to collaboratively solve the target task. Research has also examined the extent to which the wisdom of partisan crowds emerges in groups of LLM-based agents that are prompted to role-play as partisan personas (Chuang et al., 2024). And they propose a benchmark exists for evaluating their dynamics against the behavior of human groups and show the potential and limitations of LLM-based agents as a model of human collective intelligence. To reduce hallucination using reflection in multi LLM agent, Bo et al. (2024) looks at LLM-based agents with the self-reflection mechanism, where they fine-tune a shared reflector, which automatically tunes the prompts of actor models using a counterfactual PPO mechanism.

**Hallucination Detection**    Our research is therefore complementary to this line of work. There has been interesting research, including HaloScop, which uses embeddings to detect hallucination from an unlabeled corpus of text (Du et al., 2024) and Tang et al. (2024), which checks facts by training on a synthetically generated dataset. Further (Yu et al., 2024) selects the contexts and masks the attention appropriately to reduce hallucination.

# B   Proofs

## B.1   Deriving $\theta_z$ in (3)

We interpret $\theta_z(Q, \rho)$ as the probability that a uniformly random directed edge originates from a node in state $z$. Because edges are sampled by their sources, nodes with out-degree $m$ are selected with probability proportional to $m$, so the edge-source law is size-biased by $m, Q(l, m)$. Conditioning on $m$, the expected fraction of such sources that are in state $z$ is $\sum_l \rho_l(z) Q(l \mid m)$; averaging this quantity over $m$ with weights $\sum_l m Q(l, m)$ and normalizing by the total number of edges $\sum_m \sum_l m Q(l, m)$ yields the equation.

## B.2   Proof of Theorem 1

*Proof.* **(i) Existence & monotonicity.** For fixed $l$, $\theta \mapsto A_l(\theta; u)$ and $B_l(\theta; u)$ are expectations of continuous functions of $M/l$ under $\mathrm{Bin}(l, \theta)$, hence are continuous; so is $\rho_l(\cdot; u)$ and thus $\Phi(\cdot; u, Q)$, proving existence on the compact interval $[0, 1]$. Under (A1), $q \mapsto \kappa_{H,T}(\cdot, q)$ is non-decreasing and $q \mapsto \kappa_{T,H}(\cdot, q)$ is non-increasing. Coupling $M$ for $\theta_1 \leq \theta_2$ via common uniforms gives $M(\theta_1) \leq M(\theta_2)$ a.s., hence $A_l(\theta_1; u) \leq A_l(\theta_2; u)$ and $B_l(\theta_1; u) \geq B_l(\theta_2; u)$; since $\rho_l = A/(A + B)$ is increasing in $A$ and decreasing in $B$, $\rho_l$ and the weighted average $\Phi$ are non-decreasing.

**(ii) Contraction & global stability.** Write $h_H(q) := \kappa_{H,T}(u, l, q)$ and $h_T(q) := \kappa_{T,H}(u, l, q)$. By the logit form, $|h'_H(q)| = \sigma'(\Delta_H) |\partial_q \Delta_H| \leq \frac{1}{4} S_H$ and $|h'_T(q)| = \sigma'(-\Delta_T) |\partial_q \Delta_T| \leq \frac{1}{4} S_T$; thus $h_H, h_T$ are Lipschitz with constants, $L_H \leq S_H/4$, $L_T \leq S_T/4$. Under the binomial coupling, $|A_l(\theta_2; u) - A_l(\theta_1; u)| \leq L_H |\theta_2 - \theta_1|$ and similarly for $B_l$. Quotient rule for $\rho_l = A/(A + B)$ yields

$$|\rho'_l(\theta)| = \frac{|A'_l| B_l + |B'_l| A_l}{(A_l + B_l)^2} \leq \frac{\max\{L_H, L_T\}}{A_l(\theta; u) + B_l(\theta; u)} \leq \frac{\max\{S_H, S_T\}}{4 \, \eta(u)}.$$

Averaging with edge-weights gives $\sup_\theta \Phi'(\theta; u, Q) \leq \max\{S_H, S_T\}/(4 \, \eta(u)) < 1$, so $\Phi$ is a contraction. Uniqueness follows from Banach's fixed-point theorem. For $\dot{\theta} = \Phi(\theta; u, Q) - \theta$, the Lyapunov function $V(\theta) = (\theta - \theta^\star)^2$ satisfies

$$\dot{V} = 2(\theta - \theta^\star)\big(\Phi(\theta) - \Phi(\theta^\star) - (\theta - \theta^\star)\big) \leq -2\big(1 - \sup \Phi'\big)(\theta - \theta^\star)^2 < 0$$

off $\theta^\star$, so $\theta^\star$ is globally asymptotically stable.

**(iii) Comparative statics in $u$.** Let $H(\theta, u) := \theta - \Phi(\theta; u, Q)$. Since $\partial_\theta H(\theta^\star, u) = 1 - \Phi_\theta(\theta^\star; u, Q) > 0$, the implicit-function theorem gives $\frac{d\theta^\star}{du} = \frac{\Phi_u(\theta^\star; u, Q)}{1 - \Phi_\theta(\theta^\star; u, Q)}$. Differentiating $\rho_l = A/(A + B)$ in $u$ yields $\partial_u \rho_l = (A_{l,u} B_l - A_l B_{l,u})/(A_l + B_l)^2$, where $A_{l,u} = \mathbb{E}[\sigma'(\Delta_H) \, \partial_u \Delta_H] \geq 0$ and $-B_{l,u} = \mathbb{E}[\sigma'(-\Delta_T) \, \partial_u \Delta_T] \geq 0$ by (A4). Hence $\Phi_u \geq 0$, and the denominator is positive by (ii), proving monotone (strict) increase of $\theta^\star(u)$. $\square$

## B.3   Estimation Procedure

Let $\mathcal{D} = \{(u^{(m)}, l^{(m)}, i^{(m)}, j^{(m)}, w^{(m)}, z^{(m)}, z'^{(m)})\}_{m=1}^M$ collect one-step transitions observed during simulation or deployment. We estimate parameters $\theta$ of a differentiable map $r_{\theta,z}(\cdot)$ by maximizing the conditional log-likelihood

$$\max_\theta \sum_{m=1}^M \log \kappa_{z^{(m)}, z'^{(m)}}\Big(u^{(m)}, l^{(m)}, i^{(m)}, j^{(m)}; \theta\Big), \tag{5}$$

regularized as needed to prevent over-fitting when $w$ is high-dimensional. Utilities are invariant to affine transformations; we fix $r_D(\cdot) = 0$ and allow the remaining coefficients to vary freely. All reported effects are therefore in log-odds units relative to the does–not–know baseline. We also assume the Markov property and homogeneous parameters within each in-degree $l$. These simplifications keep the mean-field ODE tractable (Section 2.2); future work incorporates agent-specific random coefficients and history-dependent utilities.

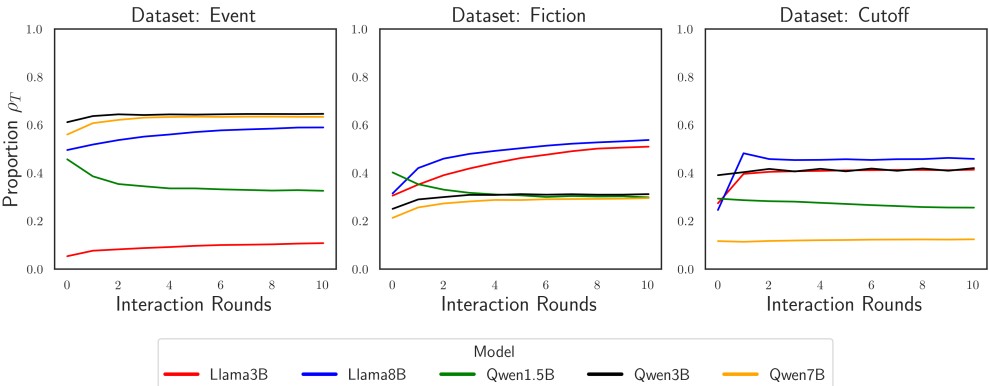

Figure 8: Different base models (Llama and Qwen with different number of parameters) exhibit different trends for the population state of truthful nodes $\boldsymbol{\rho}_T$. Based on how frequent hallucination is, the initial proportion of truthful LLMs is also different (the distribution of context and network structure is same).

## C  Experimental Details and Additional Experiments

### C.1  Experiment 8: Impact of different base models.

The base model often bounds the cognitive capabilities that the test-time scaling techniques can scale up to. We observe a similar behavior using different models in Figure 8. It can be inferred that the LLaMa-3.1 models are able to spread the correct facts better than the Qwen2.5 series models. As the size of the model scales, the eventual proportional of truthful nodes increases.

Further we perform experiments varying the communication length and number of deliberation rounds on Ministral-3 8B (mis), Gemma-3 (Team et al., 2025) and Phi-4 (Microsoft et al., 2025) and report them in Table 3. Similar to the main experiments, we observe that the proportion of truthful nodes is increasing and has diminishing returns in the control $u$. The results indicate that our insights are applicable to more model families than the ones primarily studied in the paper.

Table 3: Truthful proportion $\rho_T$ on the fiction dataset for different base models under different communication lengths and deliberation rounds.

| Control $u$ | Phi-4 | Ministral | Gemma-3 4B | Gemma-3 12B |
|---|---|---|---|---|
| Length 1 | $0.346 \pm 0.372$ | $0.325 \pm 0.391$ | $0.409 \pm 0.390$ | $\mathbf{0.427 \pm 0.307}$ |
| Length 50 | $0.476 \pm 0.331$ | $0.363 \pm 0.282$ | $0.429 \pm 0.315$ | $\mathbf{0.491 \pm 0.305}$ |
| Length 100 | $0.487 \pm 0.370$ | $0.365 \pm 0.263$ | $0.440 \pm 0.337$ | $\mathbf{0.493 \pm 0.321}$ |
| Deliberation Rounds 1 | $0.464 \pm 0.329$ | $0.422 \pm 0.294$ | $0.462 \pm 0.357$ | $\mathbf{0.469 \pm 0.305}$ |
| Deliberation Rounds 2 | $0.496 \pm 0.362$ | $0.375 \pm 0.291$ | $0.491 \pm 0.303$ | $\mathbf{0.500 \pm 0.284}$ |
| Deliberation Rounds 3 | $0.524 \pm 0.364$ | $0.434 \pm 0.304$ | $0.488 \pm 0.336$ | $\mathbf{0.535 \pm 0.321}$ |

### C.2  Synthetic Stylized Problems for Demonstrating Predictive Capabilities

We construct a synthetic dataset of five multiple-choice questions designed to study context-dependent response propagation in LLM agent networks. Each question consists of four answer choices (three specific answers plus 'I don't know') and is accompanied by three distinct types of contextual information: (1) supporting context that contains accurate information leading to the correct answer, (2) misleading context that presents plausible but incorrect information leading to a wrong answer, and (3) irrelevant context that provides no useful information for answering the question. The five questions span diverse factual domains: (Q1) 'When did the Great Library of Alexandria burn down?' (correct: 48 BCE), (Q2) 'What was the primary cause of the Bronze Age Collapse?' (correct: Climate change), (Q3) 'Who invented the

| Question ID | N=10 | N=50 | N=100 | N=500 |
|---|---|---|---|---|
| 0 | **1.00 ± 0.00** | **1.00 ± 0.00** | 1.00 ± 0.01 | 0.94 ± 0.03 |
| 1 | **1.00 ± 0.00** | **1.00 ± 0.00** | 0.91 ± 0.14 | 0.56 ± 0.03 |
| 2 | **1.00 ± 0.00** | **1.00 ± 0.00** | 0.99 ± 0.02 | 0.99 ± 0.01 |
| 3 | **0.98 ± 0.06** | **0.98 ± 0.05** | 0.96 ± 0.05 | 0.69 ± 0.04 |
| 4 | **1.00 ± 0.00** | **1.00 ± 0.00** | 0.95 ± 0.11 | 0.77 ± 0.04 |
| 5 | **1.00 ± 0.00** | **1.00 ± 0.00** | **1.00 ± 0.00** | **1.00 ± 0.00** |
| 6 | **1.00 ± 0.00** | **1.00 ± 0.00** | 1.00 ± 0.01 | 0.97 ± 0.01 |
| 7 | **1.00 ± 0.00** | **1.00 ± 0.00** | 0.80 ± 0.06 | 0.80 ± 0.03 |
| 8 | **1.00 ± 0.00** | **1.00 ± 0.00** | 0.76 ± 0.10 | 0.69 ± 0.03 |
| 9 | **1.00 ± 0.00** | **1.00 ± 0.00** | **1.00 ± 0.00** | 0.99 ± 0.01 |
| 10 | **1.00 ± 0.00** | **1.00 ± 0.00** | 0.72 ± 0.15 | 0.62 ± 0.03 |
| 11 | **1.00 ± 0.00** | **1.00 ± 0.00** | 0.83 ± 0.07 | 0.61 ± 0.02 |
| 12 | **1.00 ± 0.00** | **1.00 ± 0.00** | **1.00 ± 0.00** | 0.99 ± 0.01 |
| 13 | **1.00 ± 0.00** | **1.00 ± 0.00** | 0.79 ± 0.16 | 0.66 ± 0.04 |
| 14 | **0.98 ± 0.06** | 0.92 ± 0.12 | 0.72 ± 0.12 | 0.55 ± 0.04 |
| 15 | **1.00 ± 0.00** | 0.98 ± 0.06 | 0.91 ± 0.08 | 0.54 ± 0.02 |
| 16 | **1.00 ± 0.00** | **1.00 ± 0.00** | 0.85 ± 0.08 | 0.87 ± 0.01 |
| 17 | **1.00 ± 0.00** | **1.00 ± 0.00** | 0.72 ± 0.17 | 0.92 ± 0.01 |
| 18 | **1.00 ± 0.00** | **1.00 ± 0.00** | 0.99 ± 0.03 | 0.99 ± 0.01 |
| 19 | **1.00 ± 0.00** | **1.00 ± 0.00** | 0.81 ± 0.12 | 0.77 ± 0.01 |

Table 4: Population Truthful State $\boldsymbol{\rho}_T$ for power-law initialized network with correct context provided to influential nodes. The results indicate that for certain questions (e.g. Q15 and Q17) there is no clear trend as the number of LLMs increase from N=100 to N=500.

printing press with movable type?' (correct: Bi Sheng), (Q4) 'What percentage of human DNA is shared with bananas?' (correct: 60%), and (Q5) 'Which civilization first used zero as a number?' (correct: Ancient Indians). For each question, the misleading context provides compelling but incorrect information (e.g., attributing Gutenberg as the inventor of movable type, when Bi Sheng invented it 400 years earlier in China), while the irrelevant context offers general statements that do not aid in answering the question. This design allows us to systematically control the initial information environment and observe how different context types influence both initial agent responses and subsequent belief propagation through the network.

To establish controlled initial conditions, we assign contexts to LLMs according to a predetermined distribution: 20% of LLMs receive supporting context, 30% receive misleading context, and 50% receive irrelevant context. Each agent independently queries an LLM with their assigned context to generate an initial answer, which is then classified into one of three epistemic states: 'correct' (matching the ground truth answer), 'hallucinating' (confidently providing an incorrect answer), or 'don't know' (explicitly expressing uncertainty). Agents are arranged in a scale-free network topology generated via the Barabási-Albert (BA) preferential attachment model with parameter $m = 1$. The BA model constructs the network incrementally: starting with $m + 1$ fully connected nodes, each new node is added with $m$ edges that attach to existing nodes with probability proportional to their current degree, producing a power-law degree distribution $P(k) \sim k^{-\gamma}$ characteristic of real-world social and information networks. During each interaction timestep, a directed edge is randomly selected, and the target agent receives the source agent's answer as additional context before re-querying the LLM. This interaction protocol generates a rich dataset of state transitions that capture both the initial susceptibility to different context types and the dynamics of belief updating under peer influence, enabling empirical validation of our mean-field theoretical predictions.

### C.3 Examples from Datasets

We give 3 examples from each dataset. The first option is the correct one. The datasets are attached as supplementary material.

### C.3.1 Fiction Dataset

1. At what time did Chancellor clear the harbor mouth, prior to reaching the open Atlantic at seven in the evening?

    (a) By four o'clock in the afternoon.
    (b) Around seven in the evening.
    (c) I don't know.

2. Which national flag was lowered from the Chancellor's mast-head while threading Charleston harbor, though she was unmistakably English without colors?

    (a) The British flag.
    (b) The English flag.
    (c) I don't know.

3. According to the ship's specifications, which mast on the Chancellor did not have its base and fittings made of iron like the others?

    (a) The mizzen mast.
    (b) The main mast.
    (c) I don't know.

### C.3.2 Knowledge Cutoff Dataset

1. Which SARS-CoV-2 antigen lineage does WHO currently recommend vaccine manufacturers use for updated COVID-19 vaccines: XBB.1.5 or JN.1?

    (a) JN.1.
    (b) XBB.1.5 is currently recommended by WHO for updated COVID-19 vaccines.
    (c) I don't know.

2. What monoclonal antibody, if any, is currently authorized in the United States for COVID-19 pre-exposure prophylaxis in certain immunocompromised people?

    (a) Pemivibart (Pemgarda) is the monoclonal antibody authorized in the United States for COVID-19 pre-exposure prophylaxis in certain immunocompromised individuals.
    (b) Tixagevimab and cilgavimab, also known as Evusheld, is authorized for COVID-19 pre-exposure prophylaxis.
    (c) I don't know.

3. Have any confirmed human cases of H5N1 avian influenza linked to dairy cattle exposure been reported in the United States?

    (a) Yes, multiple confirmed human H5N1 cases in the United States have been linked to exposure to infected dairy cattle.
    (b) No confirmed human cases linked to dairy cattle exposure have been reported in the United States.
    (c) I don't know.

### C.3.3 Event Dataset

1. According to the official news, which actor vowed to repel U.S. "aggression": Caracas itself or the nation of Venezuela?

    (a) Caracas
    (b) Venezuela
    (c) I don't know.

2. Did the official news specify how many drones breached Poland before Russia's Zapad-2025 exercise, or omit any numerical count entirely?

    (a) It omitted any numerical count.

    (b) It specified 19 drones.

    (c) I don't know.

3. Which city vowed to repel 'US aggression' while diplomats fretted a state visit and apartheid-era parallels resurfaced in recent commentary?

    (a) Caracas

    (b) Havana

    (c) I don't know.

### C.4 Perturbation Analysis

To evaluate the robustness of our multi-agent system to linguistic variations, we first constructed a dataset of perturbed questions. Starting with a base set of questions from the *fiction* dataset, we employed a large language model (Gemini-2.5-Flash-Lite) to generate syntactically and lexically diverse paraphrases of each question. The generation process was guided by a structured prompting strategy, instructing the model to apply a combination of three distinct transformation types: (1) **lexical paraphrase**, replacing words with synonyms; (2) **syntactic re-framing**, altering sentence structure (e.g., active to passive voice); and (3) **indirect formulation**, converting a direct question into a polite request. By systematically applying combinations of these transformations, we created a comprehensive set of perturbed questions for each original query, ensuring that each variant preserved the core semantic intent of the original while altering its surface form.

The resulting dataset of perturbed questions was then used to conduct a series of controlled experiments on our proposed LLM network. For each original question and its set of generated perturbations, we initialized a network of LLMs with a fixed size, topology, and communication protocol. Each perturbed question was then posed to the network, initiating a multi-round communication as the rest of the paper wherein LLMs iteratively refine their answers based on information from their neighbors. To ensure the statistical significance of our results, each experiment was repeated 5 times with different random seeds. By analyzing the variance in the network's final collective answer across the different linguistic perturbations of the same underlying question, we were able to quantify the system's robustness and identify the types of linguistic variations that have the most significant impact on its collective decision-making capabilities.

