# OpenReview forum: "Collaborative QA using Interacting LLMs. Impact of Network Structure, Node Capability and Distributed Data."
_TMLR — Accepted by TMLR_

### Review · Reviewer_2jo2 · 2025-12-03

**Summary Of Contributions:**

The paper models the LLM network information spread on CQA task. It provides insight for Mean field dynamics, randomized utility model (RUM) and results validation.
Strength:
1. The problem scope is well defined.
2. The model theorems are clear in each step.
3. The experiments involves multiple stage analysis, especially for the network pattern. The insight that power law distribution network has higher variability compared to chain and tree network structures is useful.
4. It lists very good future work topics.


Weakness:
1. The experiments was conducted on base model: LLaMa-3.1-8B. It will be more interesting if the paper could involves other models' performance comparison.
2. It's better to highlight the best performed metrics in each table to make it easier to read.

**Audience:**

Yes

**Audience Explanation:**

Yes. I think this paper shows the LLM network performance with math models and experiments results. The insight regarding distribution pattern and communication overhead impact will be useful to audience to learn about information spread among collective LLMs performance.

**Broader Impact Concerns:**

The insights from the paper currently are based on 2 open models, LLaMa and Qwen. It will be more robust if the insights are proved by more based models during the experiments.

**Claims And Evidence:**

Yes

**Claims Explanation:**

1. The model algorithm is supported by precise math formula.
2. The insight generated is supported by experiment results with clear setup.
3. The motivation and citations in the literature research part is complete and right links.

**Requested Changes:**

1. It's better to add metrics highlight for best performed metrics in the table.
2. For plots like Figure 8, it's hard to differentiate 2 models performance between LLama8B and Qwen3B, suggesting to change the color or line type.

---

> ### Author Response · Authors · 2025-12-19
> **Update regarding experiment results**
>
> Dear Reviewer _2jo2_,
>
> We thank you again for your time and effort. We'd like to update you that we have added the experiments for different controls (communication length and deliberation rounds) in Table 3 of the appendix and have uploaded the updated manuscript. We experienced issues with instruction following when using the IBM Granite models and switched to the Google Gemma 3 family instead ([Gemma 3 4B](https://huggingface.co/google/gemma-3-4b-it) and [Gemma 3 12B](https://huggingface.co/google/gemma-3-12b-it)). Additionally, experiments are conducted with the Phi-4-mini-instruct and Ministral-3-8B, as mentioned earlier.
>
> Please let us know if there are any additional models or experiments that you believe would strength the claims of the contribution.
>
> Thanks again,
>
> Authors

---

### Review · Reviewer_NuyF · 2025-12-03

**Summary Of Contributions:**

This paper investigates how a network of interacting LLMs performs question answering when allowed to communicate, and when the underlying information is distributed across agents with heterogeneous contexts, computing capacities, etc.

The core contribution of the paper is a mean-field model describing the evolution of beliefs in the network of LLMs, replacing the network topology with an in-degree distribution for each node.

**Audience:**

Yes

**Audience Explanation:**

The questions that the paper tackles is definitely worthy of attention at the moment: whether connecting several language models can reduce hallucinations and what would be the compute and communication cost of doing so?
Furthermore, the simulations from the mean-field model seem mostly coherent with empirical observations, showing that the set of assumptions can somehow be realistic.

**Broader Impact Concerns:**

No broader impact is present. The research of the paper is somehow motivating using multiple LLMs to answer queries to increase true answers. Such models could also be used to study LLM network poisoning (how many contaminated nodes are needed to propagate a false belief), and justify increasing significantly the cost of each LLM use.

**Claims And Evidence:**

No

**Claims Explanation:**

The major issue is the that some claims from the paper cannot be drawn from the experiments.

The caption of table 1 states that the proportion of truthful LLMs is a concave function of the communication cost, which is not necessarily true. Concavity would be an inherent property of the function, while the diminishing return is an empirical observation. The statement that it is a concave function, and therefore has diminishing returns is reversing the empirical method: one observes diminishing return and can infer that the underlying - unknown - function is concave.

Figure 6 makes the observation that the proportion of truthful and hallucinating LLMs is
not monotonic in the number of LLMs. This observation is however not nuanced with the degree of variability of the experiments, which is only reported in Table 1. Given such a high variance (especially for Cutoff), can a conclusion be made that the proportion is indeed non monotonic or was it an artifact of noise?

"The results of Figure 7 show that the power-law distribution is the most effective in spreading factual information". The claim is not backed by the experiments. Indeed, Figure 7 only considers networks with power-law distribution of varying parameters, it seems that only Figure 4 compares network topologies and a conclusion that power-law is better is not obvious from the results.
On that note, the insight blocks at the beginning would be a good way to summarize the results, but should link to the place in the paper that the insight would be drawn from.

**Requested Changes:**

On the set of assumptions, one can assume the number of LLMs in the network will remain somehow low, in which case the mean-field approximation does not necessarily make sense. The paper should motivate why building the MF model is necessary compared to the ODE accounting for the true communication graph.

The RUM model is not motivated but simply introduced, giving a game-theoretic interpretation to an otherwise mechanistic model, the reason why LLMs would update an internal hidden state in this fashion and not another one isn't clear. The different quantities are not introduced quantitatively (is n a vector collecting the answers of a given node across time, i.e. across all communication rounds?)

Minor points
The set of edges E should be a subset of V x V, instead of an element of it.
Figure 1 would gain being vectorized.
The titles of each plot in Figure 1 should be larger or put as subcaption of the figure.
Grammar: "and placing it on less influential node can does not change"
Figure 4 attaches another meaning to the colors used for T-H-DK in all other plots. Same with figure 7.

---

> ### Author Response · Authors · 2025-12-12
>
> We thank the reviewer for their insightful comments. We appreciate the reviewer acknowledging the relevance of the research problem. We address the concerns that the reviewer has raised:
>
> > The major issue is that some claims from the paper cannot be drawn from the experiments.
> > The caption of Table 1 states that the proportion of truthful LLMs is a concave function of the communication cost, which is not necessarily true. Concavity would be an inherent property of the function, while the diminishing return is an empirical observation. The statement that it is a concave function, and therefore has diminishing returns is reverses the empirical method: one observes diminishing return and can infer that the underlying - unknown - function is concave.
>
> _Response_: We agree with the reviewer; we used the term concave as an abstraction for the phenomena that we observed. However, the reviewer is right, and we have now updated the caption to characterize the empirical observation as diminishing returns rather than convexity.
>
>
> > Figure 6 makes the observation that the proportion of truthful and hallucinating LLMs is not monotonic in the number of LLMs. This observation is however, not nuanced with the degree of variability of the experiments, which is only reported in Table 1. Given such a high variance (especially for Cutoff), can a conclusion be made that the proportion is indeed non-monotonic or was it an artifact of noise?
>
> _Response_:  We agree that the plot and result are not sufficient to back the claim that the proportion of truthful and hallucinating LLMs is not monotonic in the number of LLMs. We have removed the claim in the current form from the paper.
>
> In addition to saying something meaningful yet statistically significant, we need to run an experiment which provides evidence for perhaps a weaker yet useful claim.  The high variance in scale-free networks at small $N$ makes the claim of non-monotonicity difficult to substantiate without stricter controls. To address this, we are running a targeted experiment to isolate the effect of network size ($N$) on convergence. We test the hypothesis on a subset of questions that the proportion of truthful nodes is monotonic with respect to $N$ when the initial effective influence is held roughly constant.
>
> We are running the base experiment on the CQA task on a subset of 20 questions from the Event dataset for $N \in \{10, 50, 100, 500\}$. Crucially, to ensure systematic consistency across scales, we employ Influence-Preserved Initialization: we use rejection sampling to ensure that for every run, the initial truthful nodes collectively hold a consistent share of the network's total out-degree (normalized influence $\theta_T \approx 0.35 \pm 0.05$). This eliminates the artifact where small networks fluctuate wildly based on whether a 'hub' node is randomly assigned truth or hallucination, allowing for a fair test of monotonicity.
>
> > "The results of Figure 7 show that the power-law distribution is the most effective in spreading factual information". The claim is not backed by the experiments. Indeed, Figure 7 only considers networks with power-law distribution of varying parameters, it seems that only Figure 4 compares network topologies and a conclusion that power-law is better is not obvious from the results.
>
> _Response_:  We apologize for the incorrect reference and have corrected the caption to the correct location. Furthermore, we have qualified the claim to align with the experiments' support: “The chain has the smaller $\rho_T$ when the correct data is placed at the least influential node” and removed the previous claim.
>
>
> > On that note, the insight blocks at the beginning would be a good way to summarize the results, but should link to the place in the paper which the insight would be drawn.
>
> _Response_: Thanks for the suggestion. We have added references to the Figures and Sections in the insight blocks.

---

> ### Author Response · Authors · 2025-12-12
>
> [Continued]
>
> > On the set of assumptions, one can assume the number of LLMs in the network will remain somehow low, in which case the mean-field approximation does not necessarily make sense. The paper should motivate why building the MF model is necessary compared to the ODE accounting for the true communication graph.
>
> _Response_: We thank the reviewer for this insightful comment. We agree that for small networks ($N < 50$), the mean-field approximation is less accurate due to high variance and finite-size effects. However, our work focuses on the regime of "collective intelligence" with a larger number of agents ($N \ge 100$), where crowd-sourcing and distributed reasoning become effective.
>
> We motivate the use of the Mean-Field Dynamics (MFD) model over an ODE accounting for the exact communication graph (such as a full Markov chain) for three key reasons:
>
> 1. Combinatorial Complexity: An exact dynamical model tracking the joint state of the network would require a state space that grows exponentially with the number of agents ($|\mathcal{X}|^N$). For $N=100$, this is computationally intractable. The MFD reduces this complexity by approximating the average behavior of the population, making the derivation of the governing ODEs manageable.
>
> 2. Analytical Tractability: The MFD allows us to derive closed-form insights regarding the system's convergence. Specifically, it enables us to prove the existence, uniqueness, and global asymptotic stability of the fixed point (Theorem 1). Such analytical characterization of the equilibrium would be difficult to obtain with a high-dimensional exact model.
>
> 3. Empirical Validation: We empirically validated the approximation in Figure 2, which demonstrates that for $N=100$, the MFD model accurately predicts the dynamics of the population state with high correlation ($>0.89$) and low KL divergence.
>
> We have updated Section 2 to explicitly discuss these trade-offs and referenced the concentration inequality (Theorem 2 in [2]) which characterizes the approximation error in the finite-agent regime.
>
> [1] Li J., Zhang Q., Yu Y., Fu Q. and Ye D. “More Agents is All You Need”, TMLR, 2025.
> [2] A. Jain, V. Krishnamurthy, and Y. Zhang. Information diffusion and preferential attachment in a network of large language models, IEEE CDC, 2025, Link: https://arxiv.org/abs/2504.14438.
>
> > The RUM model is not motivated but simply introduced, giving a game-theoretic interpretation to an otherwise mechanistic model, the reason why LLMs would update an internal hidden state in this fashion and not another one isn't clear. The different quantities are not introduced quantitatively (is n a vector collecting the answers of a given node across time, i.e. across all communication rounds?)
>
> *Response*:
> Regarding the notation clarification: $n$ is not a history vector collecting answers across time, it is an indexing vector for the empirical count of the neighbours in different states. For a 3-state (T,H,D) system it would be a three dimensional vector, whose value sum up to the in-degree $l$. We have clarified this in Section 2.2.
>
> We agree that the motivation for RUM can be improved. The RUM is a macroeconomic probablistic choice model when the choice is between discrete alternatives. The model was proposed to model and analyze choices of rational agents from a population [1].
>
> While transition probabilities could indeed be estimated via simple counting (plug-in estimates) or black-box neural networks, we specifically chose RUM for two main reasons, which we have now explicitly detailed in the manuscript.
>
> 1. Interpretability: RUM grounds the model in decision theory, treating LLMs as "rational" agents that maximize a utility function involving distinct features (e.g., social consensus vs. private context). This allows us to inspect $\theta$ to quantify exactly how much weight an LLM places on its neighbors versus its own hallucination.
>
> 2. Analytical Tractability: As demonstrated in Theorem 1, the specific parametric form of the Logit model (derived from RUM under Gumbel noise) satisfies smoothness and monotonicity properties that allow us to mathematically prove the existence and uniqueness of the network's fixed point.
>
> We have improved the main text to make the motivation clear.
>
> > Minor points The set of edges E should be a subset of V x V, instead of an element of it. Figure 1 would gain being vectorized. The titles of each plot in Figure 1 should be larger or put as subcaption of the figure. Grammar: "and placing it on less influential node can does not change" Figure 4 attaches another meaning to the colors used for T-H-DK in all other plots. Same with figure 7\.
>
> *Response*: We have made the following changes:
>
> - Corrected the definition of the edges.
> - Replaced the image of Figure 1 with a vectorized PDF.
> - For Figure 4 and 7, the proportion being plotted is only the proportion of truthful nodes and we have made this clear in the figure and caption.

---

> ### Author Response · Authors · 2025-12-20
> **Update regarding experiment**
>
> Dear Reviewer NuyF,
>
> We have updated the manuscript with the additional experiment for different numbers of LLMs. We performed an experiment on a subset (20 Q/A) of the fiction dataset, where a network was initialized using a power-law degree distribution, and the correct context was provided to the influential nodes to ensure a consistent experiment as we varied the number of LLMs. The results are summarized in Table 4 in the Appendix. The results show that there are questions where there is no consistent monotonicity as the number of LLMs increases from N=100 to N=500 (specifically, Q15 and Q17). We have made the claim more exact in the manuscript:
> “We perform an additional experiment on a subset of size $20$ Q/A on the fiction dataset. We assign the correct context to the influential nodes and do $20$ independent runs for each Q/A. We report the results in Table 4 in the Appendix. There are questions (e.g. Q15 and Q17) for which there is no clear trend when increasing the number of LLMs from N=100 to N=500.”
>
>
> Further, based on your suggestion, we have also added a point in the broader impact section: “This research also forms a foundation to study network poisoning and analyze how many contaminated nodes are needed to propagate a false belief across the network.”
>
> Hopefully, this addresses, along with the rest of the rebuttal, the concerns you had regarding the claims. We’ll be happy to address any other questions or concerns you may have and make further adjustments to the manuscript based on our discussion.
>
> Thanks again for helping us improve our work.
>
> Regards,
>
> Authors

---

### Review · Reviewer_uguw · 2025-12-06

**Summary Of Contributions:**

This paper investigates the emergent behavior of networks of interacting Large Language Models (LLMs) in a Collaborative Question Answering (CQA) setting. The authors propose a hybrid theoretical framework that combines Mean-Field Dynamics (MFD) from network science to model information diffusion and a Randomized Utility Model (RUM) from economics to estimate the transition probabilities of agents changing their beliefs (between truthful, hallucinating, and "don't know" states).

The specific contributions include:
- Integrating RUM to parameterize the choice probabilities of LLMs, allowing for a data-driven estimation of how incentives and neighbors' beliefs influence an agent's transition.
- Conducting extensive experiments with 100 agents (using Llama-3 and other open-source models) on three semi-synthetic datasets (Fiction, Cutoff, and Event).
- The study demonstrates that network topology (specifically power-law distributions) and the placement of "correct" context on influential nodes significantly impact the collective convergence to truth. It also highlights that increasing the number of agents does not monotonically lead to better truthfulness, particularly when the models have strong pre-training biases (the Cutoff dataset).

**Audience:**

Yes

**Audience Explanation:**

Researchers interested in the emergent properties of "societies" of AI agents. The study of how hallucination spreads and infects a network  is crucial for understanding the risks of decentralized Agent systems.

**Broader Impact Concerns:**

The authors briefly mention that the predictive capabilities of their model could be used to "control" networks, for example, "to prevent the spread of certain sentiments".

While this is framed positively (preventing misinformation), the same mechanism describes how to optimize a network for propaganda or targeted disinformation campaigns. The paper explicitly models how to manipulate the "truth" by placing specific contexts on influential nodes.

Therefore, the authors is recommended to explicitly acknowledge the dual-use nature of "optimizing information diffusion." what is the risk of bad actors using these findings (e.g., power-law topology exploitation) to engineer echo chambers or amplify hallucinations?

**Claims And Evidence:**

Yes

**Claims Explanation:**

1. The MFD model accurately predicts the dynamics of the LLM network. Figure 2
2. Network topology significantly affects the spread of truth; specifically, Power-Law networks are more effective than random networks. Figure 4 and 7.
3. Placing "correct" context on influential nodes improves the collective accuracy. Figure 4 and 5
4. Test-time compute (deliberation) scales well to networks. Table 3 and Figure 3

**Requested Changes:**

1. Reconcile Sequential Theory vs. Parallel Experiment: The discrepancy between the sequential nature of the ODE derivation and the parallel execution of the experiments  must be addressed rigorously.

The MFD derivation explicitly assumes sequential interactions where one edge is sampled at a time. However, the experiments involve 100 LLMs interacting in parallel for 10 rounds. While the authors acknowledge this "simplifying assumption" for the model, they do not sufficiently analyze how this discrepancy affects the theoretical guarantees (e.g., the contraction mapping proofs in Theorem 1)

2. Figure 6 shows that for the "Cutoff" dataset, increasing agents from 10 to 100 decreases the proportion of truthful nodes. Can you provide a concrete hypothesis or analysis in the text explaining why this happens. Is it because the "hallucination" is the dominant prior in the pre-training data, and social pressure forces agents to revert to the training mean?

3. The reference of Table 3 is incorrect.

4. Readability. The current manuscript is textually dense and occasionally disjointed, e.g., section 2 put a focus on the WHAT but lacks the WHY, the transition from the heavy derivations in Section 2 to the experimental setup in Section 3 is abrupt.

5. Notations in Equation 1 and 4, and connections to the experiments. Equation 1 does not define new variables such as u. explicitly mention in the experiments what represent the variables in your previous equations.

---

> ### Author Response · Authors · 2025-12-17
>
> We thank the reviewer for acknowledging the paper's strengths and contributions, including the modeling of extensive experimentation, the use of RUM to estimate transition probabilities, and the mapping of insights to the experiments.  We’d like to reply to the concerns raised by the reviewer:
>
> > The MFD derivation explicitly assumes sequential interactions where one edge is sampled at a time. However, the experiments involve 100 LLMs interacting in parallel for 10 rounds. While the authors acknowledge this "simplifying assumption" for the model, they do not sufficiently analyze how this discrepancy affects the theoretical guarantees (e.g., the contraction mapping proofs in Theorem 1)
>
> Response: The MFD model used in information diffusion is a continuous-time ODE model, where for each time interval $[t,t+dt]$, a node is sampled for interaction. This sequential sampling of the edge allows us to write the ODE (equation 1 in the paper) in terms of the net change in the proportion of different states of the population. We do this to derive a clean set of equations characterizing the fixed point. The reviewer is correct in pointing out that there is a discrepancy from the parallel interactions of the experiments (as we acknowledge in the system model). In general, deriving the dynamics for parallel interactions is difficult, and the sequential assumption is a standard modeling assumption to obtain a continuous-time mean-field limit, although interactions are parallel in nature in practical real-life systems  [1].
>
> However, particularly the results of Theorem 1, which characterize the properties of the fixed point, hold true for parallel interactions, since the fixed point exists and is unique; therefore, the nature of the interactions only affects the transient dynamics, not the fixed point. Specifically, the map $\Phi$ is defined purely by the static probability of an agent adopting the truth given the current neighbor statistics, making its functional form independent of whether updates occur one by one or all at once. Consequently, the derivative bound ($\Phi' < 1$) derived in Theorem 1 is an intrinsic property of the map itself, guaranteeing contraction for both the discrete parallel iteration and the continuous sequential dynamics.
>
> We have added an explanation regarding the same in Section 2.4.
>
> [1] Jackson, M. O., & López-Pintado, D. (2013). Diffusion and Contagion in Networks with Heterogeneous Agents and Homophily. Network Science, 1(1), 49-67.
>
> > Figure 6 shows that for the "Cutoff" dataset, increasing agents from 10 to 100 decreases the proportion of truthful nodes. Can you provide a concrete hypothesis or analysis in the text explaining why this happens. Is it because the "hallucination" is the dominant prior in the pre-training data, and social pressure forces agents to revert to the training mean?
>
> Response: As reviewer Nyuf also points out, the results in the previous state of the paper were not statistically significant, and we performed the experiment on a subset of $20$ questions of the Fiction dataset for $20$ runs each. We observe that for $N=10$ agents, the standard deviation is still too high to make any conclusive statement. We will update the paper and provide with a concrete hypothesis for the observed behaviour if possible.
>
> > The reference of Table 3 is incorrect
>
> Response: We have corrected the reference in the updated manuscript.
>
> > Readability. The current manuscript is textually dense and occasionally disjointed, e.g., section 2 put a focus on the WHAT but lacks the WHY, the transition from the heavy derivations in Section 2 to the experimental setup in Section 3 is abrupt.
>
> Response: We have added motivation for the mean field dynamics used to model the information diffusion, and RUM used to model the transition probabilities in Section 2. We have also added a paragraph at the end of Section 2 to improve the transition to Section 3.
>
> > Notations in Equation 1 and 4, and connections to the experiments. Equation 1 does not define new variables such as u. explicitly mention in the experiments what represent the variables in your previous equations.
>
> Response: We have improved the notation of the equation, ensured all the variables are well-defined and the experiments explicitly reference the variable names along with the symbols. Although symbols like $u$ were defined in the system model, we have redefined them here after the equation they appear in (equation 1, in the case of $u$).

---

> > ### Author Response · Authors · 2025-12-17
> >
> > [CONTINUED]
> >
> >
> > > Broad Impact Concerns:
> > > The authors briefly mention that the predictive capabilities of their model could be used to "control" networks, for example, "to prevent the spread of certain sentiments".
> > > While this is framed positively (preventing misinformation), the same mechanism describes how to optimize a network for propaganda or targeted disinformation campaigns. The paper explicitly models how to manipulate the "truth" by placing specific contexts on influential nodes.
> > > Therefore, the authors is recommended to explicitly acknowledge the dual-use nature of "optimizing information diffusion." what is the risk of bad actors using these findings (e.g., power-law topology exploitation) to engineer echo chambers or amplify hallucinations?
> >
> >
> > Response: We appreciate the reviewer's suggestion. Indeed, a discussion on broader impact can improve the exposition of the paper and inform future research. Based on the reviewer's suggestion, we have added the following paragraph in the Section 4,
> >
> > "Similar to other research that aimed at reducing undesirable effects like hallucinations, the research presented in this paper can also be used for adversarial or malicious purposes. Particularly, one can optimize the network structure, incentives, and communication protocols to spread hallucination, triggering content, and misinformation. Bad actors can misuse techniques from network science and optimization to engineer echo chambers and amplify certain sentiments, which, paired with sycophancy, can be very detrimental to human-AI interactions on the internet. Therefore, there is more research needed to make LLMs and their networks robust, safe, and reliable.
> > This research also forms a foundation to study network poisoning and analyze how many contaminated nodes are needed to propagate a false belief across the network."

---

### Author Response · Authors · 2025-12-20
**Brief Summary of Response by Authors and Revision to the Manuscript**

Dear Reviewers and Action Editor,

We thank you all for your time and effort in scientifically and rigorously evaluating our work. We are glad that all the reviewers found the problem setting, approach, and insights interesting. Specifically, this work lays the foundation for addressing questions like: what are the risks in societies of agents (reviewer NuyF), what is the effect of connecting several large language models on hallucinations and the corresponding computational tradeoff (reviewer uguy), and how information spreads in a network of LLMs (reviewer 2jo2). Reviewer NuyF and uguy appreciated that our insights our supported by experiments.

The reviewers also raised certain concerns, which we summarize below along with our response and the changes we made to the paper.

[1] Claims and Results

Reviewer NuyF pointed out that "concavity" implies an inherent property of the function, whereas "diminishing returns" is an empirical observation 1. We agree and have updated the caption of Table 1 to characterize the observation as diminishing returns. Regarding the claim that the proportion of truthful and hallucinating LLMs is not monotonic with respect to $N$ (Reviewer uguw, NuyF), we acknowledged that the high variance in scale-free networks at small $N$ made the initial claim difficult to substantiate. As promised, we performed the targeted experiment to ensure consistency across scales ($N=10, 50, 100, 500$). The results confirm that there are questions where increasing the number of nodes does not necessarily increase or decrease the proportion of truthful LLMs in the last iterate.

[2] Explanation and Motivation for RUM and MFD

In response to Reviewer NuyF’s request to better motivate the modeling choices, we have expanded Section 2\. We explain that we chose the Mean-Field Dynamics (MFD) model over an exact dynamical model (like a full Markov chain) because the latter has a state space that grows exponentially with the number of agents ($|\\mathcal{X}|^N$), making it computationally intractable for $N=100$. MFD provides analytical tractability, allowing us to prove the existence and uniqueness of the fixed point (Theorem 1). Regarding the Randomized Utility Model (RUM), we clarified that we chose it over black-box methods for interpretability, it grounds the LLM behavior in decision theory and analytical tractability, as the Logit model properties allow us to mathematically prove the network's fixed point properties.

[3] Sequential versus Parallel

Reviewer uguw noted the discrepancy between the sequential nature of our MFD derivation and the parallel execution of the experiments. We have added an explanation in Section 2.4, clarifying that while the sequential assumption is standard for obtaining a continuous-time mean-field limit, the results of Theorem 1 (characterizing the fixed point) also hold true for parallel interactions. The contraction mapping guarantees derived in the theorem rely on the static probability map, meaning the nature of the interaction (sequential vs. parallel) affects the transient dynamics but not the existence or uniqueness of the fixed point.

[4] Experiments on other models

Reviewer 2jo2 suggested involving other models beyond Llama-3 to make the insights more robust. We have significantly expanded our experimental validation in Appendix C.1. In addition to Llama-3, we have now included experiments with the Ministral-3-8B, Phi-4-mini-instruct, and the Google Gemma 3 family (Gemma 3 4B and Gemma 3 12B). We also updated Figure 8 with an improved color scheme to better differentiate between model performances.

[5] Improving Writing

We have addressed the readability and structural concerns raised by Reviewer uguw and NuyF. We added a "Broader Impact" discussion in Section 4 to explicitly acknowledge the dual-use nature of optimizing information diffusion, including risks related to echo chambers and misinformation amplification. We improved the transition between the theoretical derivations in Section 2 and the experimental setup in Section 3\. Furthermore, we linked the insight blocks at the beginning of the paper to the specific Figures and Sections they summarize, as suggested by Reviewer NuyF.

[6] Typos and Notations

We have corrected the typos and notation issues identified by the reviewers. Specifically, we explicitly defined variables such as $u$ and $n$ (clarifying $n$ as an indexing vector for neighbor counts, not a history vector. We corrected the definition of the edge set $E$ to be a subset of $V \\times V$. We have also corrected the reference to Table 3 and replaced Figure 1 with a clearer, vectorized version.

We thank the reviewers again and hope that we were able to address the concerns they had. We’d be glad to answer any other questions and concerns that the reviewers have. Looking forward to a productive discussion.

Regards,

Authors

---

> ### Author Response · Authors · 2026-01-12
> **Follow-up on Revisions and Responses**
>
> Dear Reviewers and Action Editor,
>
> We thank you for your time and effort.
> We are writing to follow up on our previous responses and the revised manuscript posted on December 20th. We believe we have addressed the concerns raised by all reviewers, specifically:
> - Reviewer NuyF: We conducted the requested targeted experiments to isolate the effect of network size ($N$) on convergence and updated the claims regarding monotonicity.
> - Reviewer 2jo2: We significantly expanded the experimental validation to include additional model families (Gemma 3, Phi-4-mini, Ministral-3-8B) to ensure the robustness of our insights.
> - Reviewer uguw: We clarified the theoretical distinction between sequential and parallel interactions and expanded the discussion on broader impacts.
>
> Please let us know if there are any outstanding questions or if you require further clarifications. We are happy to engage in additional discussion to ensure the paper meets the venue's standards.Thank you again for your time and guidance in improving our work.
>
> Best regards,
>
> Authors

---

### Decision · Action_Editor_TW7U · 2026-02-03

**Recommendation:** Accept with minor revision

**Additional Comments:**

Various minor elements are still pending:
- the formatting of some new paragraphs could be improved.
- at the end of 2.4, "Discrepancy between sequential MFD and parallel experimentation" should be a paragraph head.
- the sentence added at the start of 3.2 misses a reference.
- the vertical spacing of Table 2 is too narrow below the first row.
- the authors need of course to change blue text back to black and remove strike-through text, but also do a general, thorough proofreading for errors/typos etc.

Other points that could be improved:

- The formula at the end of page 5 for $\mathbf{F}^l$ is negative whenever $z_1=z_2$. However, this quantity denotes the transition rate between different states and should therefore be a non-negative real number between 0 and 1. Perhaps the authors meant 1 - the negative quantity?
- The authors mainly experiment with power-law networks. They did not try other types of networks though, such as small-world networks, regular networks, exponential networks, etc. It would be nice for them to elaborate on the reasons behind this choice, or alternatively include more network types in their evaluation.

**Audience:**

Yes

**Audience Explanation:**

The problem setting is clearly defined and relevant to current research on multi-agent LLM systems. Overall, the paper makes a contribution to the growing literature on multi-agent LLM collaboration. The methodology is clear, and so is the analysis, whereas the findings are likely to be of interest to both researchers and practitioners working on distributed and agent based language model systems.

**Claims And Evidence:**

Yes

**Claims Explanation:**

The paper proposes a mean-field analysis of the dynamic of belief propagation throughout a network of LLMs, and uses a randomized utility model (RUM) to represent the inner dynamic of each agent.

The specific contributions include:
- Integrating RUM to parameterize the choice probabilities of LLMs, allowing for a data-driven estimation of how incentives and neighbors' beliefs influence an agent's transition.
- Conducting extensive experiments with 100 agents (using Llama-3 and other open-source models) on three semi-synthetic datasets (Fiction, Cutoff, and Event).
- The study demonstrates that network topology (specifically power-law distributions) and the placement of "correct" context on influential nodes significantly impact the collective convergence to truth. It also highlights that increasing the number of agents does not monotonically lead to better truthfulness, particularly when the models have strong pre-training biases (the Cutoff dataset).

On the other hand, as Reviewer NuyF pointed out, <<various minor elements are still pending: the formatting of some new paragraphs could be improved, at the end of 2.4, "Discrepancy between sequential MFD and parallel experimentation" should be a paragraph head. The sentence added at the start of 3.2 misses a reference. The vertical spacing of Table 2 is too narrow below the first row.>>

Other points that could be improved:
- The formula at the end of page 5 for $\mathbf{F}^l$ is negative whenever $z_1=z_2$. However, this quantity denotes the transition rate between different states and should therefore be a non-negative real number between 0 and 1. Perhaps the authors meant 1 - the negative quantity?
- The authors did not include an ablation with simple counting or black-box neural networks. This may have been a nice comparison.
- The authors mainly experiment with power-law networks. They did not try other types of networks though, such as small-world networks, regular networks, exponential networks, etc. It would be nice for them to elaborate on the reasons behind this choice, or alternatively include more network types in their evaluation.

---

> ### Author Response · Authors · 2026-02-23
>
> Dear Action Editor,
>
> Thank you for your comments. We are happy to know that the paper has been accepted with minor revisions. We are deligthed that you and the reviewers found the paper interesting and relevant to research in multi-agent LLM collaboration.
> We have made the requested changes to the manuscript and have uploaded the camera ready revision.
>
> Specifically we have made the following changes:
>
> - We have improved the formatting of the new paragraphs:
>   - At the end of 2.4, "Discrepancy between sequential MFD and parallel experimentation" is a paragraph head.
>   - Missing references have been added
>   - The vertical spacing of Table 2 has been fixed
>   - We have removed the blue text and have fixed the erros and typos.
>
> - We have added a clarification for the transition rates between different states based on the discussion with the action editor.
>
> - We have added reasoning for why we mainly consider power-law networks in the experiment section.